# FutureOmni: Evaluating Future Forecasting from Omni-Modal Context for Multimodal LLMs

Qian Chen [1]   Jinlan Fu [1]   Changsong Li [1 2]   Min Zhang [3]   See-Kiong Ng [4]   Xipeng Qiu [1 2]

## Abstract

Although Multimodal Large Language Models (MLLMs) demonstrate strong omni-modal perception, their ability to forecast future events from audio-visual cues remains largely unexplored, as existing benchmarks focus mainly on retrospective understanding. To bridge this gap, we introduce FutureOmni, the first benchmark designed to evaluate omni-modal future forecasting from audio-visual environments. The evaluated models are required to perform cross-modal causal and temporal reasoning, as well as effectively leverage internal knowledge to predict future events. FutureOmni is constructed via a scalable LLM-assisted, human-in-the-loop pipeline and contains 919 videos and 1,034 multiple-choice QA pairs across 8 primary domains. Evaluations on 13 omni-modal and 7 video-only models show that current systems struggle with audio-visual future prediction, particularly in speech-heavy scenarios, with the best accuracy of 64.8% achieved by Gemini 3 Flash. To mitigate this limitation, we curate a 7K-sample instruction-tuning dataset and propose an Omni-Modal Future Forecasting (OFF) training strategy. Evaluations on FutureOmni along with standard audio-visual and video-only benchmarks show that OFF improves future forecasting performance and generalization. Code and data are available at https://github.com/OpenMOSS/FutureOmni.

## 1. Introduction

Multimodal Large Language Models (MLLMs) have demonstrated remarkable capabilities in audio-video understanding (Google, 2025; Xu et al., 2025b). To access these capabilities, community has established a suite of omni-modal benchmarks designed to evaluate synergistic video-audio understanding (Geng et al., 2025; Zhou et al., 2025c; Chao et al., 2025; Li et al., 2025a; Yang et al., 2025). Recent suites, such as WorldSense (Hong et al., 2025) and Daily-Omni (Zhou et al., 2025b), have expanded this scope, testing MLLMs on complex tasks ranging from event captioning to temporal grounding and holistic question answering. These benchmarks have successfully driven progress in retrospective reasoning (Nguyen et al., 2025; Chowdhury et al., 2025; Chen et al., 2026), enabling models to accurately describe and analyze events have occurred within videos.

Although extensive research has focused on retrospective reasoning, predicting future events is equally critical in real-world applications. For example, in autonomous driving, systems must integrate auditory cues (e.g., honking from nearby vehicles) with visual information (e.g., pedestrian positions) to anticipate future world states and make timely safety decisions. Some prior works have explored future forecasting. FutureBench (Wang et al., 2025), Forecast-Bench (Karger et al., 2025), FutureX (Zeng et al., 2025), and MIRAI (Ye et al., 2024) predict real-world future events and evaluate language-based LLMs considering only textual modality, and require periodic benchmark updates to prevent data leakage. VLEP (Lei et al., 2020), IntentQA (Li et al., 2023), and MM-Forecast (Li et al., 2024a) extend future prediction to vision and language modalities. However, the auditory modality, despite its importance for future reasoning, has been largely overlooked. As a result, the capability of multimodal LLMs to perform future forecasting from joint audio-visual inputs remains insufficiently studied.

In this paper, we introduce FutureOmni, the first comprehensive benchmark for evaluating Multimodal LLMs on future event forecasting under an omni-modal context, with a focus on cross-modal causal forecasting, temporal reasoning, and the effective use of internal knowledge. As illustrated in Fig. 1, an evaluated MLLM is required to select the correct future event based on the omni-modal context, which in this case consists of audio and video modalities. FutureOmni is constructed using a scalable AI-assisted, human-in-the-loop pipeline to enable efficient dataset construction while maintaining high data quality. It comprises 919 videos and 1,034

[1]Fudan Univeriusity, Shanghai, China [2]Shanghai Innovation Institute, Shanghai, China [3]Harbin Institute of Technology, Shenzhen, China [4]National University of Singapore. Correspondence to: Jinlan Fu <jinlanjonna@gmail.com>.

*Proceedings of the $43^{rd}$ International Conference on Machine Learning*, Seoul, South Korea. PMLR 306, 2026. Copyright 2026 by the author(s).

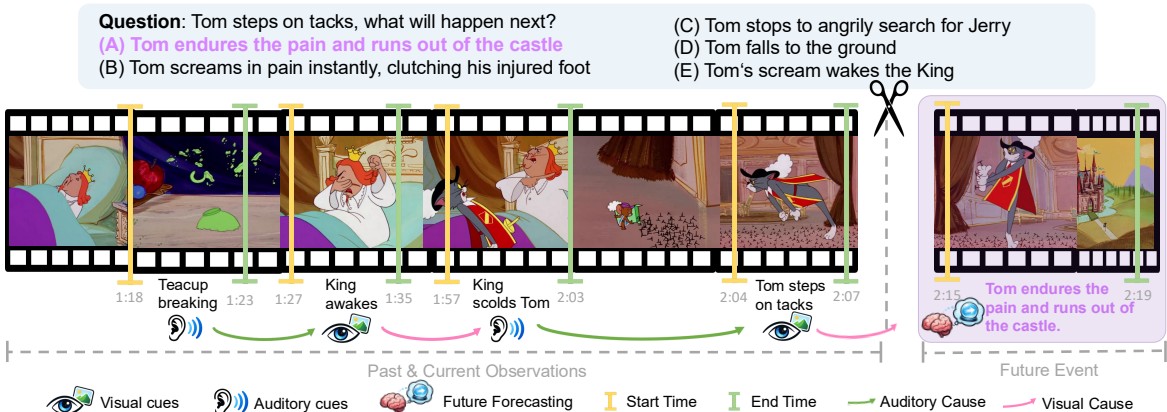

*Figure 1.* An Example from FutureOmni illustrating the Omni-modal Future Prediction task. The green and pink arrows denote consequences induced by auditory and visual cues, respectively.

multiple-choice QA pairs spanning 8 primary domains. To ensure comprehensive evaluation, we collect videos with diverse audio types (speech, environmental sounds, and music) and durations up to 20 minutes. To prevent potential shortcut learning, we design four types of adversarial distractors. By introducing visual-only and audio-only conflicts, as well as delayed and reverse-causal options, we require MLLMs to perform genuine cross-modal reasoning for future forecasting.

**Experiments and Findings**. We conduct extensive experiments on widely used MLLMs, including both omni-modal and video-centric models, covering proprietary and open-source systems, as shown in Fig. 2. The results reveal a key limitation: *(F1) existing omni-modal and video-centric MLLMs struggle to accurately forecast future events under omni-modal contexts, with the best performance reaching only 64.8%, achieved by Gemini 3 Flash.* To address this issue, we construct a high-quality instruction-tuning dataset, namely, FutureOmni-7K, and propose an Omni-Modal Future Forecasting (OFF) method. Evaluations on FutureOmni, popular audio-visual benchmarks (e.g., WorldSense, DailyOmni), and video-only benchmarks (e.g., Video-MME) show that *(F2) OFF substantially improves the performance of open-source models on our benchmark and enhances their out-of-domain generalization.* Furthermore, attention-score visualizations indicate that *(F3) OFF improves the model's ability to identify critical keyframes, leading to better generalization and reasoning performance.*

Our main contributions are as follows:

(1) We introduce FutureOmni, the first benchmark for evaluating the future forecasting ability of MLLMs under omni-modal contexts. It contains 919 videos and 1,034 QAs across 8 domains, filling a key gap in omni-modal reasoning evaluation.

(2) We conduct extensive evaluation and analysis on 20

MLLMs. The results show that both omni-modal and video-only models exhibit limited future forecasting ability, with even the best proprietary model achieving only 64.8%, revealing substantial room for improvement.

(3) To enhance omni-modal LLMs, we annotate 7K instruction-tuning samples and propose an Omni-modal Future Forecasting (OFF) method. Results demonstrate that OFF improves future forecasting and generalization ability, as supported by attention visualizations.

## 2. Related Work

**MLLMs**. Video LLMs using temporal visual inputs for video understanding (Zhang et al., 2023; Li et al., 2024b) become paradigms. In parallel, LLMs have been augmented with auditory perception by integrating pretrained audio encoders, enabling both speech and non-speech audio understanding (Radford et al., 2023; Tang et al., 2024; Chu et al., 2024; Hu et al., 2024). Recently, Omnimodal Large Language Models capable of processing and reasoning across text, vision, and audio simultaneously become popular. Proprietary state-of-the-art models, such as Gemini 3 Flash (Google, 2025) have showcased remarkable abilities in handling long-context, interleaved audio-visual inputs. Within the open-source landscape, the dual-tower paradigm, which processes visual and acoustic signals via distinct encoders, has gained traction. Prominent examples include Qwen2.5-Omni (Xu et al., 2025a) and video-SALMONN 2 (Tang et al., 2025a) have substantially advanced the integration of audio modalities into video LLMs.

**Multimodal Benchmarks**. For omni-modal evaluation, recent benchmarks such as WorldSense (Hong et al., 2025) and Daily-Omni (Zhou et al., 2025b) further incorporate audio cues into visual QA. Nevertheless, existing datasets emphasize retrospective reasoning, leaving omnimodal future prediction underexplored.

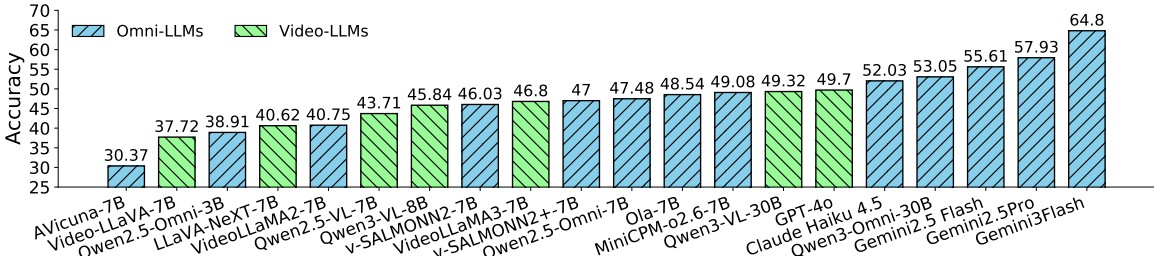

*Figure 2.* Overall scores on FutureOmni.

Conversely, regarding the future prediction task, early benchmarks like VLEP (Lei et al., 2020) and IntentQA (Li et al., 2023) require models to predict future actions or anticipate long-term goals based on the current context. Despite their value, they are predominantly vision-centric. They typically function with the audio track muted or disregarded, failing to capture scenarios where sound acts as the primary precursor to a future event. Consequently, there is a lack of benchmarks that demand omni-modal perception and causal future reasoning, a gap that FutureOmni aims to bridge.

## 3. The FutureOmni Benchmark

### 3.1. Audio Coordinated Video Selection

For our evaluation, low-quality videos are characterized by short duration, static scene changes, or audio serving a decorative role. Firstly, we collect approximately 18K YouTube videos ranging from 30 seconds to 20 minutes and apply an audio coordinated video filtering strategy. Videos with limited scene change are removed by computing frame-level visual similarity between adjacent frames and discarding samples whose average inter-frame similarity is higher than 70%. Inspired by AVoCaDO (Chen et al., 2025), we propose an audio intervention strategy to filter out videos with weak audio-visual correlations or merely decorative audio tracks, which calculates the semantic similarity gap between captions generated with and without audio input. We select videos with larger similarity drop indicating audio dependency. We choose UGCVideoCaptioner (Wu et al., 2025) for efficiency, and we only select the top-50% videos for subsequent annotation. Finally, we obtain a subset of 9K videos. Experiment details are in Appendix A.7.

### 3.2. Audio-Visual Temporal Localization and Calibration

After filtering, the next challenge is to locate and describe events within the videos. Previous research typically relied on open-source grounding models (Ren et al., 2024); however, these models either lack the precision required for dense video caption (Zhang et al., 2025b; Geng et al., 2025), or they ignore the audio input. In this paper, we leverage the advanced multimodal capabilities of Gemini 2.5 Flash (Google, 2025) to implement grounding. Specifically, we first instruct the model to perform a comprehensive scan of the video to identify plot-relevant events while ignoring trivial or static background occurrences. We require the model to generate precise timestamps (in MM:SS format) that tightly bound the duration of each event.

**Time Boundary Checking**. To validate the precision of these boundaries, following LongVALE (Geng et al., 2025), we compute Mel-frequency cepstral coefficients (MFCCs) at the start and end points; since valid event transitions typically correlate with acoustic discontinuities, we verify whether MFCC differences at these time points exceed a pre-defined threshold of 2.0.

**Audio Fulfilling**. Following boundary validation, we enrich the temporal segments through prompting Gemini 2.5 Flash to identify and annotate specific acoustic cues—such as dialogue, sound effects, or background music—that occur synchronously with the visual content, ensuring that audio events are captured alongside the visual action.

### 3.3. Audio-Visual QA Construction

With the dense, omni-modal event timeline established in the previous stage, the final step involves extracting logical cause-and-effect pairs from sequential events. As illustrated in Fig. 3, this process consists of two key phases: Causal Pair Discovery and Dual-Stage Verification.

**Causal Pair Discovery**. Mere temporal succession does not imply causality. To identify events where the future is logically predictable from the past, we employ DeepSeek-V3.2 (DeepSeek-AI et al., 2025) to analyze adjacent event segments. To ensure the reliability of the prediction task, we strictly limit the temporal gap between the premise and the future event to a maximum of 30 seconds. We feed the model the chronologically ordered descriptions and instruct it to determine if the subsequent event is a direct logical consequence of the former. The model is required to explicitly output three components: **Premise** Event, **Target** Event and **Rationale**, which is a logical explanation bridging the gap. To explicitly mine pairs driven by acoustic

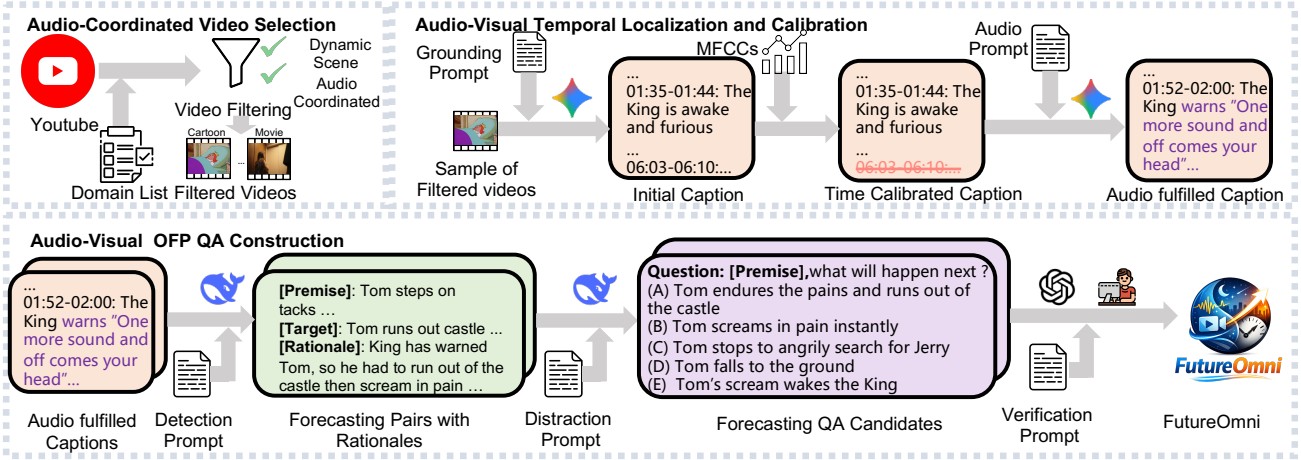

*Figure 3.* The pipeline of our FutureOmni.

*Table 1.* Comparison of FutureOmni with other representative video and audio-visual benchmarks. A and M in **Annotation** indicate automatic and manual annotation. **FF Patterns** denotes Future Forecasting patterns, including Thematic Montage (T), Causal (C), and Routine Sequences (R). **FF QAs** represents the amount of future forecasting QAs. **New Video** denotes whether videos are newly collected. ✗ represents partially collected (52.8%) in FutureBench.

| Benchmarks | Videos | Avg.Duration(s) | QAs | Annotation | w. Audio | FF Patterns | FF QAs | New Video |
|---|---|---|---|---|---|---|---|---|
| VLEP (2020) | 528 | 33.1 | 4,192 | M | ✗ | T,C,R | 4,192 | ✓ |
| IntentQA (2023) | 567 | 46.4 | 2,134 | A+M | ✗ | C | 503 | ✗ |
| FutureBench (2025) | 866 | 43.1 | 1,056 | A+M | ✗ | T,C,R | 1,056 | ✗ |
| AVUT (2025) | 2,662 | 67.8 | 13,774 | A+M | ✓ | ✗ | ✗ | ✓ |
| LongVALE (2025) | 8,400 | 235.0 | ✗ | A+M | ✓ | ✗ | ✗ | ✗ |
| DailyOmni (2025b) | 684 | 42.8 | 1197 | A+M | ✓ | ✗ | ✗ | ✗ |
| JointAVBench (2025) | 1,046 | 97.2 | 2,853 | A+M | ✓ | ✗ | ✗ | ✗ |
| OmniVideoBench (2025a) | 628 | 384.2 | 1,000 | A+M | ✓ | ✗ | ✗ | ✓ |
| WorldSense (2025) | 1,662 | 141.1 | 3,172 | M | ✓ | ✗ | ✗ | ✗ |
| FutureOmni | 919 | 163.5 | 1,034 | A+M | ✓ | T,C,R | 1,034 | ✓ |

cues, we instruct the model to score the audio causal factor for each candidate pair. Specifically, the model assigns a contribution score on a scale of 0 to 2, where 0 represents no influence, 1 indicates decoration, and 2 denotes causality. Furthermore, the model classifies the audio factor into three distinct categories: Speech, Sound, and Music.

**QA Construction**. Unlike prior event prediction benchmarks (Lei et al., 2020; Li et al., 2023) that primarily construct distractors based on visual similarity, we propose four novel distractor types to rigorously evaluate the omnimodal reasoning abilities of MLLMs: i) **Visual-only Perception**, which is visually plausible given the video context but is explicitly contradicted by the audio modality. This targets models that fail to integrate auditory cues into their reasoning process. ii) **Audio-only Perception**, which aligns semantically with the speech or sound events but describes visual actions that do not occur or mismatch the visual scene. This challenges models that over-rely on the audio transcript or sound processing while neglecting visual verification. iii) **Delayed**. It describes valid past events from the video that

occur before the premise. This evaluates the model's temporal precision ability. iv) **Reverse-Causal**. It describes the antecedent or cause of the premise event rather than its effect. This tests the model's understanding of the directional arrow of time.

**Dual-Stage Verification** To mitigate the ambiguity of our data, we implement a dual-stage verification strategy. Candidate QAs are submitted to GPT-4o (OpenAI, 2024) for automated logical validation first. Then we conduct human verification to check quality. Details are in Appendix B.

### 3.4. Dataset Statistics

Tab. 1 and Fig. 4 present a comprehensive comparison between FutureOmni and other representative benchmarks. Our dataset comprises 919 high-quality videos and 1,034 QA pairs. While recent omnimodal benchmarks like World-Sense and DailyOmni incorporate audio, they focus largely on retrospective perception or captioning, ignoring the future forecasting task. FutureOmni is the first to bridge this gap, dedicating 100% of its samples (1,034 QAs) to Fu-

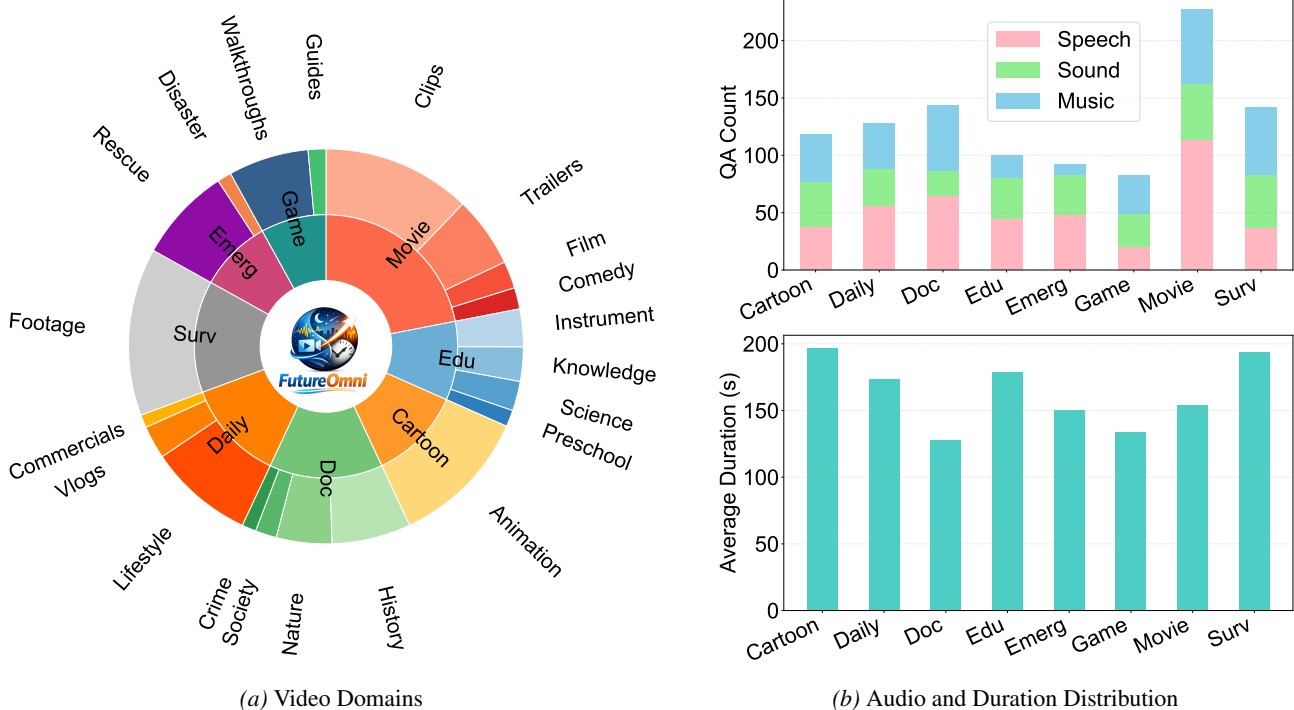

*(a)* Video Domains

*(b)* Audio and Duration Distribution

*Figure 4.* (a) Hierarchical distribution of 8 primary video domains and 21 fine-grained sub-categories. Surv: Surveillance, Daily: Dailylife, Edu: Education, Emerg: Emergency. (b) Composition of audio modalities and average video duration.

ture Forecasting. Moreover, the average video duration in FutureOmni is 163.5 seconds, significantly longer than traditional prediction datasets like VLEP (33.1s) and FutureBench (43.1s). Unlike benchmarks limited to specific logic types (e.g., IntentQA focuses only on Causal), FutureOmni covers a diverse spectrum of reasoning patterns, including **Thematic Montage (T)** (Semantic Coherence or atmospheric continuity), **Causal (C)** (Physical or logical cause-and-effect), and **Routine Sequences (R)** (Habitual or procedural order). This ensures a holistic evaluation of a model's predictive capabilities.

## 4. Experiments

### 4.1. Settings

We evaluate a broad spectrum of representative MLLMs on FutureOmni, categorized into three groups:(1) open-source video-audio MLLMs, such as MiniCPM-o 2.6 (Yao et al., 2024) , video-SALMONN 2 (Tang et al., 2025a) , Ola-7B (Liu et al., 2025b) , Qwen2.5-Omni (Xu et al., 2025a) and Qwen3-Omni (Xu et al., 2025b). (2) open-source video MLLMs, such as VideoLLaMA3 (Zhang et al., 2025a) and Qwen3-VL (Bai et al., 2025a). (3) proprietary MLLMs, such as Claude Haiku 4.5 (Anthropic, 2025) , Gemini 2.5 Flash, Pro (Google, 2025) and Gemini 3 Flash (Google, 2025). All models are evaluated using their official implementations. Performance is measured by direct comparison

between outputs and ground-truth annotations.

### 4.2. Results on FutureOmni

Tab. 2 reports the performance of representative MLLMs. We evaluate 20 models across three categories: Open-Source Omni-MLLMs, Open-Source Video MLLMs, and Proprietary MLLMs. The results yield several insightful observations.

- *Open-source Omni-LLMs still lag behind proprietary models.* Our results indicate proprietary Omni-LLMs, namely Gemini 2.5 Pro and Gemini 3 Flash, achieve an average accuracy of approximately 61%, whereas the strongest open-source Omni-LLM attains only 53%. This persistent performance gap suggests that open-source models with joint audio–visual processing capabilities remain underexplored and offer considerable potential for further improvement.

- *Video-only LLMs consistently underperform Omni-LLMs due to their inability to leverage audio cues.* Even competitive proprietary video-only models, such as GPT-4o, achieve a maximum accuracy of 49.70%, which is lower than that of open-source Omni-LLMs such as Qwen3-Omni. The gap is even larger for other video-only models, highlighting the importance of audio–visual integration in future event prediction.

*Table 2.* Overall performance on FutureOmni. Edu:Education, Emerg: Emergency, Surv: Surveillance, Daily: Dailylife, Doc:Documentary. Models with a gray background indicate proprietary systems.

| Methods | Size | Cartoon | Edu | Emerg | Surv | Daily | Movie | Game | Doc | Avg |
|---|---|---|---|---|---|---|---|---|---|---|
| *Video-Audio MLLMs* | | | | | | | | | | |
| AVicuna (Tang et al. 2025b) | 7B | 31.62 | 39.00 | 26.09 | 35.21 | 32.81 | 28.19 | 33.73 | 20.83 | 30.37 |
| VideoLLaMA2 (Cheng et al. 2024) | 7B | 43.59 | 47.00 | 29.35 | 53.52 | 40.62 | 32.60 | 57.83 | 31.94 | 40.75 |
| Qwen2.5-Omni (Xu et al. 2025a) | 3B | 37.61 | 51.00 | 29.35 | 57.75 | 35.94 | 32.16 | 51.81 | 25.00 | 38.91 |
| video-SALMONN 2 (Tang et al. 2025a) | 7B | 43.59 | 55.00 | 39.13 | 57.04 | 48.44 | 40.97 | 57.83 | 34.72 | 46.03 |
| video-SALMONN 2+ (Tang et al. 2025a) | 7B | 50.43 | 61.00 | 39.13 | 55.63 | 52.34 | 40.09 | 54.22 | 33.33 | 47.00 |
| Qwen2.5-Omni (Xu et al. 2025a) | 7B | 47.86 | 55.00 | 35.87 | 59.86 | 48.44 | 40.09 | 61.45 | 40.28 | 47.48 |
| Ola (Liu et al. 2025b) | 7B | 44.44 | 62.00 | 42.39 | 64.08 | 47.66 | 41.41 | 59.04 | 37.50 | 48.54 |
| MiniCPM-o 2.6 (Yao et al. 2024) | 8B | 48.72 | 63.00 | 43.48 | 59.15 | 50.00 | 41.85 | 62.65 | 36.11 | 49.08 |
| Qwen3-Omni (Xu et al. 2025b) | 30B | 52.94 | 68.00 | 32.88 | 62.71 | 59.05 | 45.60 | 62.65 | 49.25 | 53.05 |
| Claude Haiku 4.5 (Anthropic 2025) | - | 55.08 | 66.00 | 44.57 | 57.04 | 51.56 | 48.90 | 57.83 | 41.67 | 52.03 |
| Gemini 2.5 Flash (Google 2025) | - | 50.85 | 70.00 | 47.83 | 59.15 | 58.59 | 51.54 | 60.24 | 50.00 | 55.61 |
| Gemini 2.5 Pro (Google, 2025) | - | 49.15 | 75.00 | 54.35 | 69.01 | 62.50 | 51.54 | 65.06 | 46.53 | 57.93 |
| Gemini 3 Flash (Google, 2025) | - | 62.71 | 75.00 | 58.70 | 80.28 | 68.75 | 59.03 | 65.06 | 53.47 | **64.80** |
| *Video MLLMs* | | | | | | | | | | |
| Video-LLaVA (Lin et al. 2024) | 7B | 39.32 | 47.00 | 33.70 | 41.55 | 42.19 | 32.16 | 44.58 | 29.86 | 37.72 |
| LLaVA-NeXT (Zhang et al. 2024) | 7B | 43.59 | 49.00 | 31.52 | 49.30 | 35.94 | 38.33 | 50.60 | 31.94 | 40.62 |
| Qwen2.5-VL (Bai et al. 2025b) | 7B | 43.59 | 58.00 | 30.43 | 52.82 | 48.44 | 37.00 | 53.01 | 34.72 | 43.71 |
| Qwen3-VL (Bai et al. 2025a) | 8B | 39.32 | 64.00 | 34.78 | 58.45 | 48.44 | 38.33 | 57.83 | 36.11 | 45.84 |
| VideoLLaMA3 (Zhang et al. 2025a) | 7B | 42.74 | 59.00 | 33.70 | 58.16 | 42.97 | 43.61 | 67.47 | 35.66 | 46.80 |
| Qwen3-VL (Bai et al. 2025a) | 30B | 41.88 | 66.00 | 43.48 | 59.15 | 53.12 | 41.85 | 61.45 | 39.58 | 49.32 |
| GPT-4o (OpenAI 2024) | - | 44.06 | 65.00 | 34.78 | 57.74 | 52.34 | 50.22 | 51.80 | 36.11 | **49.70** |

**Breakdown Results** (1) *Performance varies significantly across domains.* Models generally perform better in Game and Dailylife categories (e.g., Qwen3-Omni scores 62.65% on Game), likely due to the predictable nature of game physics and common daily routines. Conversely, the Documentary (Doc) and Emergency domains prove the most difficult, with average scores dropping to the 20-40% range (e.g., AVicuna scores only 20.83% on Doc). We hypothesize that Documentaries often rely on complex narration (speech) to explain visual phenomena, while Emergency scenarios require rapid processing of chaotic audio-visual cues (e.g., sirens, screams), posing a severe test for current models' synergistic reasoning abilities.

(2) *A Contextual Cold Start phenomenon is observed across all Omni-LLMs.* As illustrated in Fig. 5b, all models struggle most with the shortest duration across four intervals, achieving the lowest scores (e.g., Qwen3-Omni with 34.90% and Gemini 3 Flash with 40.78%). Performance peaks in the medium duration range ([2,4) min) before slightly dipping for long videos. This could be attributed to the fact that future forecasting requires sufficient historical context to establish a prediction, while short videos often lack the necessary narrative.

(3) *Speech is consistently the most challenging modality.* As presented in Fig. 5a, Qwen3-Omni shows an approximately 10% gap between Music (57.54%) and Speech (47.99%). Even Gemini 3 Flash scores 60.52% on Speech versus 68.31% on Music. This suggests that speech requires

high-level linguistic decoding and semantic alignment with visual cues, posing a greater barrier than interpreting atmospheric music or distinct sound events.

**Modality Ablation** To quantify the specific contribution of each modality to future prediction, we conduct a modality ablation study on four strongest open-source omni-modal models: Ola, Qwen2.5-Omni, MiniCPM-o 2.6 and Qwen3-Omni (Details are in Appendix A.3). The results in Tab. 3 lead to three key conclusions. (1) *The full omni-modal setting (A+V) consistently yields the highest performance.* For instance, Qwen2.5-Omni achieves 47.48% with both modalities, but drops significantly to 42.50% when provided with only video (V) or only audio (A). This substantial performance gap (approx. 5%) empirically validates the core premise of FutureOmni: accurate future prediction relies on the synergistic integration of visual dynamics and acoustic cues, rather than on either modality in isolation. (2) *While supplementing video with text-based information—such as Subtitles (V+Subtitle) or detailed Captions (V+Caption)—improves performance over the video-only baseline, it still falls short of the full A+V setting.* For example, Ola scores 48.54% with raw audio but only 46.95% with subtitles. This suggests that the raw audio signal contains rich, non-verbal latent information (e.g., emotional tone, environmental atmosphere, urgency) that cannot be fully captured by textual transcription alone. (3) *The performance of omni-modal models on Audio-only (A) and Video-only (V) is strikingly similar.* This indicates that the

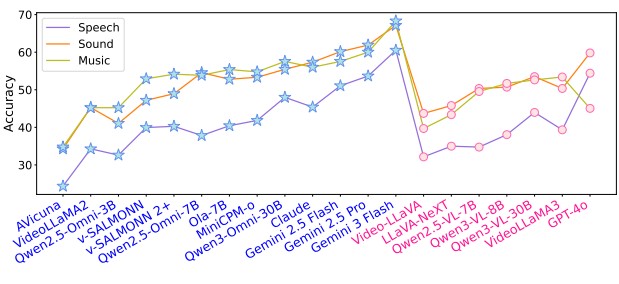

(a) Results on three audio types.

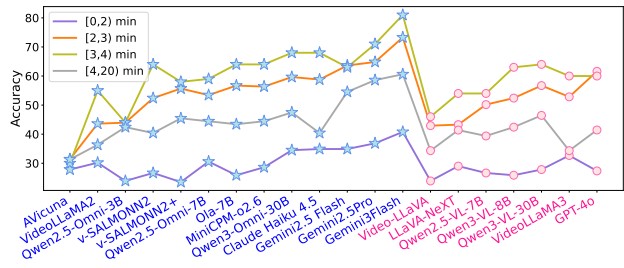

(b) Results on four duration intervals.

*Figure 5.* Fine-grained results on audio types (a) and duration intervals (b).

dataset is well-balanced; the models cannot shortcut the task by relying solely on visual pattern matching or audio classification. To achieve the state-of-the-art results seen in the A+V column, the model must genuinely perform cross-modal reasoning.

*Table 3.* Ablation Results. A:audio, V:video, S:subtitle, C:caption.

| Methods | A+V | V | V+S | A | A+C |
|---|---|---|---|---|---|
| Qwen3-Omni (30B) | 53.05 | 51.50↓ | 52.76↓ | 50.92↓ | 50.34↓ |
| MiniCPM-o 2.6 (8B) | 48.54 | 48.25↓ | 49.80↑ | 48.93↑ | 48.54 |
| Ola (7B) | 48.54 | 43.27↓ | 46.95↓ | 46.47↓ | 46.85↓ |
| Qwen2.5-Omni (7B) | 47.48 | 42.50↓ | 43.85↓ | 42.50↓ | 44.24↓ |

**Error Analysis**  We choose 318 failure cases from Gemini 3 Flash into four categories between knowledge deficits and reasoning failures. Details are in Appendix A.4. The distribution in Fig. 6 reveals three insights: (1)*Visual Perception is the primary bottleneck*. The majority of errors (51.6%) stem from Video Perception Errors. This indicates that despite strong general capabilities, the SOTA model still struggles to capture the fine-grained visual dynamics. (2) *The Synergistic Reasoning Gap*. A substantial portion (30.8%) are Audio-Video Joint Reasoning Failures. In these cases, the model perceives individual modalities but fails to synthesize them logically (e.g., failing to link a visual action with its corresponding sound effect) to derive the future, validating the omni-modal challenge of our benchmark. (3) *Reasoning over Knowledge*. Notably, Lack of Knowledge accounts for a negligible 2.5% errors. This confirms that current MLLMs possess sufficient world knowledge; the performance gap on FutureOmni is driven by limitations in dynamic perceptions and complex causal reasoning rather than a lack of factual data.

*Table 4.* Performance on audios with OFF on FutureOmni.

| Methods | Speech | Sound | Music | Avg |
|---|---|---|---|---|
| Qwen2.5-Omni | 37.83 | 54.55 | 53.85 | 47.48 |
| +OFF | **47.75** | 47.55 | 50.46 | **48.51** |
| video-SALMONN 2 | 39.95 | 47.20 | 52.92 | 46.03 |
| +OFF | **44.68** | **54.39** | 52.62 | **49.90** |
| Ola | 40.43 | 52.80 | 55.38 | 48.54 |
| +OFF | **42.55** | **53.50** | **57.23** | **50.19** |

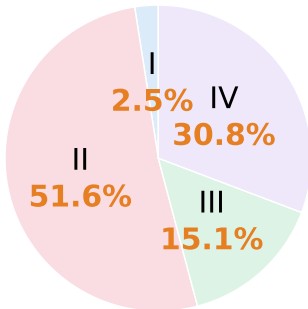

*Figure 6.* Error Distribution.I: Lack of Knowledge. II: Video Perception Error. III: Audio Perception Error. IV: Audio-Video Joint Reasoning Failure.

# 5. Omnimodal Future Forecasting: From Prediction to Generalization

As demonstrated in our experiments, current MLLMs exhibit distinct limitations in omni-modal future forecasting. To bridge this gap, we curate a high-quality instruction tuning dataset, FutureOmni-7K, and propose the Omni-modal Future Forecasting (OFF) training method. The dataset comprises 7,761 instruction-following samples derived from the high-quality filtered videos (Sec. 3.1) that are explicitly excluded from the evaluation benchmark, ensuring a strict disjoint split to prevent data leakage. Crucially, during training, we only **input video frames and audio segments occurring before the target event**. This constraint prevents the model from simply memorizing future frames and forces it to perform genuine predictive reasoning based on the historical context. Furthermore, each sample is enriched with a detailed rationale that explicates the causal logic connecting the audio-visual premise to the future outcome, encouraging the model to internalize the underlying mechanisms of forecasting rather than merely fitting the answer distribution.

## 5.1. Main Experiment

**Settings**  We conduct experiments on three representative open-source omni-modal models: Qwen2.5-Omni-7B, Ola-7B, and video-SALMONN 2-7B. To ensure computational efficiency, we use LoRA (Hu et al., 2022) for fine-tuning.

*Table 5.* General Capability Analysis Results. **Bold** values indicate **best performance within each method**.

| Methods | Audio-Visual Bench | | | | Video-only Bench | |
|---|---|---|---|---|---|---|
| | WorldSense | DailyOmni | JointAVBench | OmniVideoBench | Video-MME | MLVU |
| Qwen2.5-Omni | 37.67 | 45.69 | 59.30 | 30.70 | 53.77 | 54.00 |
| +OFF | **40.22** | **49.03** | **60.88** | **31.70** | **55.51** | **54.37** |
| video SALMONN 2 | 48.29 | 65.13 | 60.21 | 34.90 | 61.40 | 68.00 |
| +OFF | **48.77** | **65.80** | **61.16** | **35.40** | 61.25 | 67.86 |
| Ola | 44.07 | 53.04 | 50.29 | 35.50 | 48.00 | 51.93 |
| +OFF | 44.10 | **53.63** | **51.13** | **36.90** | 48.07 | **52.53** |

During the training process, we keep the visual and audio encoders frozen and only update text backbones. The learning rate is set to 1e-5, and the models are trained for 1 epoch. We freeze the vision and audio encoder when training. All other hyperparameters and configurations remain consistent with the official training scripts of the respective models. Other details are in Appendix A.1.

**Results** Tab. 4 demonstrates the effectiveness of OFF. All models exhibit consistent gains, with video-SALMONN 2 achieving the largest overall increase of +3.87%. Most notably, the training significantly boosts performance in the challenging Speech category; Qwen2.5-Omni witnesses a substantial leap of nearly 10%. This confirms that our instruction tuning effectively enhances the models' ability to interpret complex acoustic cues, particularly dialogues.

### 5.2. General Capability Analysis

To investigate whether the future prediction capability acquired from FutureOmni-7K can transfer to general domains, we evaluate our fine-tuned models on a suite of out-of-domain benchmarks.

**Settings** Specifically, we selected four representative omni-modal benchmarks: WorldSense, DailyOmni, JointAVBench, and OmniVideoBench, to test audio-visual synergy. Additionally, to assess impacts on pure visual understanding, we included two Video-only benchmarks: Video-MME (Fu et al., 2025) and MLVU (Zhou et al., 2025a).

**Results** The results in Tab. 5 demonstrate the generalization ability of OFF, with training solely on future prediction leads to consistent improvements across multiple omni-modal QA tasks not relevant with the forecasting. For instance, Qwen2.5-Omni achieves notable gains on WorldSense (+2.55%) and DailyOmni (+3.34%). Remarkably, these benefits extend even to video-only benchmarks. Qwen2.5-Omni shows clear improvements on Video-MME (53.77% to 55.51%) and MLVU (54.00% to 54.37%).

**Attention Visualization** To investigate the mechanism of this generalization, we analyze the internal attention distribu-

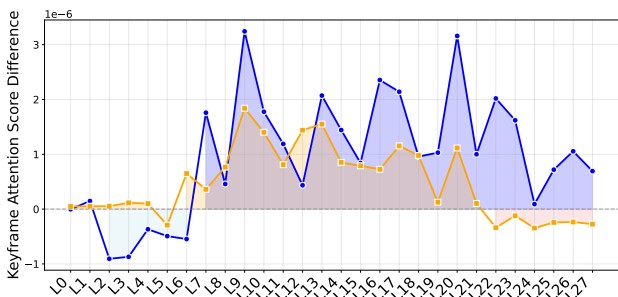

*Figure 7.* Attention score difference visualization. The **blue** represents the attention difference for video keyframes, while the **orange** represents audio keyframes.

tion of Qwen2.5-Omni before and after training. We choose LongVALE for this experiment due to the well-annotated video and audio keyframes. We propose a metric, called *Keyframe Attention Score Difference*, which measures the shift in attention magnitude assigned to ground-truth video and audio keyframes across transformer layers. As illustrated in Fig. 7, the results uncover the following pattern beneficial to the generalization: *Active Information Seeking*. The trained model (positive values) pays more attention to both Video (**Blue**) and Audio (**Orange**) keyframes in these critical layers. For instance, at Layer 9 and Layer 20, the model's focus on visual cues intensifies dramatically. Simultaneously, the audio attention (Orange) shows a consistent elevation across the middle layers (L8-L17). More quantitative results are in App. C.

## Conclusion

In this work, we introduce FutureOmni, the first comprehensive benchmarks dedicated to evaluating the Omni-modal Future Prediction capabilities of MLLMs. By establishing a rigorous human-in-the-loop data construction pipeline, we curate a high-quality dataset that strictly demands audio-visual reasoning. Our extensive evaluation reveals that current MLLMs struggle with OFP, particularly in speech-dense scenarios. To address this deficiency, we construct a rationale-enhanced instruction-tuning dataset, FutureOmni-7K and propose an Omni-Modal Future Forecasting training strategy not to sharpen cross-modal future prediction abilities, but to enhance performance across general tasks.

## Acknowledgments

This work was supported by the National Natural Science Foundation of China (No. U24B20181 and 62525602). This research/project is supported by the National Research Foundation, Singapore under its National Large Language Models Funding Initiative (AISG Award No: AISG-NMLP-2024-002). Any opinions, findings and conclusions or recommendations expressed in this material are those of the author(s) and do not reflect the views of National Research Foundation, Singapore.

## Impact Statement

We emphasize that FutureOmni is intended for research purposes to diagnose and improve model reasoning. We encourage the community to carefully monitor the deployment of predictive MLLMs to ensure they are used ethically and transparently.

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

*Table 6.* Comparison of MLLMs across different audio types and video durations.

| Models | Audio Type | | | Video Duration | | | | Avg |
|---|---|---|---|---|---|---|---|---|
| | Speech | Sound | Music | [0,2)min | [2,3)min | [3,4)min | [4,20)min | |
| *Video-Audio LLMs* | | | | | | | | |
| AVicuna | 24.35 | 34.27 | 34.77 | 27.84 | 31.38 | 30.00 | 31.31 | 30.37 |
| VideoLLaMA2 | 34.28 | 45.26 | 45.23 | 30.19 | 43.62 | 55.00 | 36.36 | 40.75 |
| Qwen2.5-Omni-3B | 32.62 | 41.05 | 45.23 | 23.92 | 43.96 | 44.00 | 42.42 | 38.91 |
| video SALMONN 2 7B | 39.95 | 47.20 | 52.92 | 26.66 | 52.41 | 64.00 | 40.40 | 46.03 |
| video SALMONN 2+ 7B | 40.28 | 48.95 | 54.15 | 23.53 | 55.69 | 58.00 | 45.45 | 47.00 |
| Qwen2.5-Omni-7B | 37.83 | 54.55 | 53.85 | 30.58 | 53.44 | 59.00 | 44.44 | 47.48 |
| Ola-7B | 40.43 | 52.80 | 55.38 | 25.88 | 56.72 | 64.00 | 43.43 | 48.54 |
| MiniCPM-o 2.6 8 | 41.84 | 53.33 | 54.77 | 28.62 | 56.37 | 64.00 | 44.44 | 49.08 |
| Qwen3-Omni-30B | 47.99 | 55.44 | 57.54 | 34.50 | 59.65 | 68.00 | 47.47 | 53.05 |
| Claude Haiku 4.5 | 45.39 | 57.34 | 56.00 | 34.90 | 58.79 | 68.00 | 40.40 | 52.03 |
| Gemini 2.5 Flash | 51.06 | 60.14 | 57.54 | 34.90 | 63.62 | 63.00 | 54.55 | 55.61 |
| Gemini 2.5 Pro | 53.66 | 61.89 | 60.00 | 36.86 | 64.83 | 71.00 | 58.59 | 57.93 |
| Gemini 3 Flash | 60.52 | 67.13 | 68.31 | 40.78 | 73.28 | 81.00 | 60.61 | 64.80 |
| *Video-LLMs* | | | | | | | | |
| Video-LLaVA | 32.15 | 43.71 | 39.69 | 23.92 | 42.93 | 46.00 | 34.34 | 37.72 |
| LLaVA-NeXT | 34.99 | 45.80 | 43.38 | 29.02 | 43.28 | 54.00 | 41.41 | 40.62 |
| Qwen2.5-VL-7B | 34.75 | 50.35 | 49.54 | 26.66 | 50.17 | 54.00 | 39.39 | 43.71 |
| Qwen3-VL-8B | 38.06 | 50.70 | 51.69 | 25.88 | 52.41 | 63.00 | 42.42 | 45.84 |
| Qwen3-VL-30B | 43.97 | 53.50 | 52.62 | 27.84 | 56.72 | 64.00 | 46.46 | 49.32 |
| VideoLLaMA3 | 39.34 | 50.35 | 53.40 | 32.67 | 52.84 | 60.00 | 34.34 | 46.80 |
| GPT-4o | 54.41 | 59.80 | 45.05 | 27.37 | 61.61 | 60.00 | 41.44 | 49.70 |

# A. Appendix

## A.1. Training and inference details with FutureOmni-7K

All experiments are conducted on 8 H200s. For Qwen2.5 Omni-7B, we use LlamaFactory(Zheng et al., 2024) and set LoRA rank 64. We use AdamW and cosine scheduler. We set batch size as 1. For Ola and video SALMONN 2, we follow the training scripts released by the official repos. For evaluation on general benchmarks, we use vLLM to speed up. A training example is listed as follows:

---

**Instruction Tuning Example (OFF Strategy)**

**User:** `<video><audio>`
Given the premise event: "Instructions appear on how to download and install the VISION WELL app, showing QR code and store download buttons", which event is its most direct conclusion?

A. The app shows a 'Connection succeeded' screen with green checkmarks, followed by instructions to name the device and select its location
B. Screenshots show the app's network configuration steps where users generate a QR code for the camera to scan
C. The video concludes with a 'Technical Assistance' screen, displaying a toll-free number and a website for support
D. A diagram highlights the power button on the camera, with instructions to fully charge the battery, then long-press the power button for 3 seconds to turn it on
E. The video shows screenshots of the app interface, instructing new users to create an account or sign in, and to connect their phone to the Wi-Fi network

---

**Assistant: Rationale:** The initial instruction to download the app (cause) logically precedes and enables the later network configuration step (effect) where the app is used to generate a QR code. Without first installing the app, the user cannot access the network configuration features shown later in the process.

**Therefore the answer is: B**

---

## A.2. Attention Visualization Details

We sample 1000 samples from LongVALE-test. For attention analysis, since the sparsity of the video and audio modality (Tao et al., 2025), we only calculate top-25% scores for each layer.

## A.3. Modality Ablation Details

Subtitles were transcribed from the audio track using Voxtral-Mini-3B (Liu et al., 2025a). Captions were generated using VideoChat-R1-7B-caption (Li et al., 2025b), prompted to provide a detailed visual description of the video content.

## A.4. Error Analysis Details

**Error Definitions**    Audio Perception Error: The correct prediction relies heavily on a specific sound (speech, event, or music) that the model clearly ignored or hallucinated. Indicator: The visual context alone is insufficient or misleading, and the model failed because it missed the acoustic cue (e.g., a doorbell ringing off-screen). Video Perception Error: The correct prediction relies on a visual detail (object, text, or action) that the model failed to recognize. Lack of Knowledge: The model likely perceived the sensory data correctly but lacked the external world knowledge, physics, or domain expertise required to predict the outcome. Audio-Video Joint Reasoning Failure: The model correctly perceives both the visual and audio elements individually but fails to combine them logically to derive the causal future.

## A.5. Statistics of FutureOmni

FutureOmni is organized into eight major category groups, comprising 21 fine-grained subcategories in total, covering a wide range of video domains and reasoning demands.

**Cartoon (1 subcategory)**: This group includes Animation, focusing on stylized visual content and narrative understanding in animated videos.

**Education (4 subcategories)**: This group consists of Instrument, Knowledge, Science, and Preschool(Kids' Education), emphasizing instructional clarity, factual reasoning, and multimodal alignment in educational scenarios.

**Emergency (2 subcategories)**: This category includes Rescue and Disaster, targeting safety-critical situations that require accurate temporal reasoning and event understanding.

**Surveillance (1 subcategory)**: This category covers Footage(Police Footage), focusing on real-world monitoring scenarios with complex and rapidly evolving visual events.

**Dailylife (3 subcategories)**: This group contains Commercials, Vlogs (Travel Vlogs), and Lifestyle(Lifestyle Vlogs), representing diverse user-generated content with informal narration and varied filming styles.

**Movie (4 subcategories)**: This category includes Clips (TV Scene Clips), Trailers (TV Trailers), Film (Film Analysis), and Comedy(Comedy Skits), emphasizing narrative coherence, character interactions, and cinematic understanding.

**Game (2 subcategories)**: This category comprises Guides, Walkthroughs.

**Documentary (4 subcategories)**: This category includes Crime, Society, Nature and History.

Overall, this taxonomy ensures broad coverage across entertainment, education, real-world documentation, and safety-critical analysis, enabling comprehensive evaluation of multimodal models under diverse and challenging video understanding scenarios.

## A.6. Video Filtering Details

To ensure the dataset contains dynamic visual content, we first filter out static videos. We sample frames at 1 FPS and extract visual feature embeddings using CLIP-ViT-B/32. We define the visual stability metric as the cosine similarity between adjacent frame embeddings; videos with an average inter-frame similarity higher than 0.7 are discarded.

## A.7. UGCVideoCaptioner Experiment

To ensure strong audio-visual correlation, we apply an audio intervention strategy. We first generate two captions for each video using UGCVideoCaptioner: one conditioned on both audio and visual inputs, and another conditioned on visual input only. We then encode each caption using Sentence-BERT (all-MiniLM-L6-v2) and compute their semantic similarity via cosine similarity between normalized embeddings. Similarity scores are computed in GPU batches for efficiency. Videos are ranked by similarity, and we retain the top 50% with the largest scores, where caption differences indicate a stronger influence of audio information. The prompt used for caption generation is provided below:

---

**UGCVideoCaptioner**

You are given a short video with both audio and visual content. Write a detailed and coherent paragraph that naturally integrates all modalities. Your description should include:

(1) the primary scene and background setting;

(2) key characters or objects and their actions or interactions;

(3) significant audio cues such as voices, background music, sound effects, and their emotional tone;

(4) any on-screen text (OCR) and its role in the video context; and

(5) the overall theme or purpose of the video. Ensure the output is a fluent and objective paragraph, not a bullet-point list, and captures the video's content in a human-like, narrative style.

---

*Table 7.* Performance changes on categories training with FutureOmni-7K .

| Methods | Cartoon | Edu | Emerg | Surv | Daily | Movie | Game | Doc | Avg |
|---|---|---|---|---|---|---|---|---|---|
| Qwen2.5-Omni | 47.86 | 55.00 | 35.87 | 59.86 | 48.44 | 40.09 | 61.45 | 40.28 | 47.48 |
| +OFF | 50.92 | 58.82 | 49.33 | 41.10 | 65.25 | 55.24 | 44.56 | 62.65 | 48.51 |
| video-SALMONN 2 | 43.59 | 55.00 | 39.13 | 57.04 | 48.44 | 40.97 | 57.83 | 34.72 | 46.03 |
| +OFF | 42.59 | 60.00 | 50.00 | 61.97 | 55.46 | 40.97 | 63.85 | 37.50 | 49.90 |
| Ola | 44.44 | 62.00 | 42.39 | 64.08 | 47.66 | 41.41 | 59.04 | 37.50 | 48.54 |
| +OFF | 44.44 | 59.00 | 42.39 | 66.20 | 51.56 | 44.49 | 60.24 | 40.28 | 50.19 |

## B. Human Verification Details

To ensure the reliability of the generated QAs, we implement a rigorous human review process involving two expert annotators specializing in multimodal reasoning. Each candidate QA pair, after passing the automated GPT-4o check, is independently reviewed based on three strict criteria: (1) logical uniqueness of the correct answer, (2) necessity of audio-visual integration, and (3) plausibility of distractors. To quantify the consistency of our annotation, we compute the Inter-Annotator Agreement (IAA) using Fleiss' Kappa, achieving a score of 0.73, indicating substantial agreement. Furthermore, approximately 20% of the candidate samples are rejected or required manual revision during this phase, ensuring that only instances with unambiguous ground truth and strong causal dependencies are retained in the final FutureOmni benchmark.

## C.  Why Future Forcasting Transfers

We argue that OFF improves transferability by explicitly training models to identify causal triggers and reason over temporal progression, thereby encouraging a structured understanding of timelines instead of relying on shallow correlations. As shown in Tab. 9, OFF consistently improves performance not only on FutureOmni, but also across fine-grained temporal reasoning benchmarks, including WorldSense (Temporal Prediction and Temporal Localization), JointAVBench (Plot Temporal Grounding), and OmniVideoBench (Temporal Understanding). These gains quantitatively demonstrate that the transferability of OFF is not a superficial, generalized boost, but rather a targeted enhancement of the precise cognitive capability required for future forecasting: temporal reasoning.

To further validate this claim, we introduce a strong baseline, where models are fine-tuned on generic video captions of comparable scale to FutureOmni-7K. Empirical results from Tab. 8 show that OFF consistently outperforms this caption-

based baseline across different architectures (Qwen2.5-Omni, video-SALMONN 2, and Ola) and multiple benchmarks. The advantage is especially pronounced on tasks requiring anticipation and temporal prediction. For example, on FutureOmni, OFF improves video-SALMONN 2 from 46.89 to 49.90 and Ola from 48.02 to 50.19, substantially surpassing standard caption supervision. These findings indicate that the observed improvements are not merely a byproduct of additional training data. Instead, they stem from the unique nature of the future forecasting objective, which forces models to internalize causal relationships and temporal progression, capabilities that generic descriptive SFT data fail to effectively cultivate.

*Table 8.* Comparison between OFF and SFT across FutureOmni and general audio-visual benchmarks.

| Model | FutureOmni | | WorldSense | | DailyOmni | | JointAVBench | | OmniVideoBench | |
|---|---|---|---|---|---|---|---|---|---|---|
| | OFF | SFT | OFF | SFT | OFF | SFT | OFF | SFT | OFF | SFT |
| Qwen2.5-Omni | **48.51** | 47.09 | **40.22** | 38.49 | **49.03** | 47.70 | **60.88** | 59.58 | **31.70** | 31.20 |
| video SALMONN2 | **49.90** | 46.89 | **48.77** | 48.01 | **65.80** | 64.97 | **61.16** | 60.43 | **35.40** | 34.33 |
| Ola | **50.19** | 48.02 | **44.19** | 44.01 | **53.63** | 53.05 | **51.13** | 50.11 | **36.90** | 35.09 |

*Table 9.* Comparison between OFF and SFT on fine-grained temporal reasoning tasks.

| Benchmark | Fine-grained Task Type | OFF | SFT |
|---|---|---|---|
| WorldSense | Temporal Localization | **1.78%** | -0.59% |
| | Temporal Prediction | **5.45%** | 1.82% |
| JointAVBench | PTG | **6.62%** | 2.21% |
| OmniVideoBench | Temporal Understanding | **2.92%** | 1.46% |

# D. Prompt

**Audio-Visual Temporal Localization**

Analyze the provided video for pairs of events that have a causal relationship that crosses modalities. A cross-modality causal relationship exists when an event from one modality (video or audio) makes a subsequent event from the other modality predictable. For the purpose of this task, "audio event" refers to non-speech sounds (e.g., music, sound effects, ambient noise).

**Instructions**

a. Focus on Plot-Relevant Events: Prioritize causal pairs that are essential for understanding the narrative or plot development of the video. These are events that drive the story forward, rather than simple, everyday occurrences.

b. Avoid Commonsense Causal Pairs: Do not list simple, predictable cause-and-effect relationships that are based on basic commonsense knowledge. For example, avoid pairs like:

"has_causal: audio to video" - a doorbell sound followed by a person opening a door.

"has_causal: video to audio" - a person striking a match followed by a scratching sound.

"has_causal: audio to video" - an elevator 'ding' sound followed by the elevator doors opening.

**Output Format**

For each such causal event pair, provide the modality of the premise and the conclusion, along with the details of each event. Your output must follow this exact format:

has_causal: [premise_modality] to [conclusion_modality]

premise_event: [premise_start_time]:[premise_end_time], [premise_event_description]

conclusion_event: [conclusion_start_time]:[conclusion_end_time], [conclusion_event_description]

**Example Output**

has_causal: video to audio

premise_event: 01:22:01:25, A person's hand presses a large, red button.

conclusion_event: 01:25:01:28, A loud mechanical whirring sound is heard.

has_causal: audio to video

premise_event: 02:45:02:48, The audio plays a dramatic musical crescendo.

conclusion_event: 02:48:02:51, The video shows a character jumping from a building.

Please analyze this video and identify all cross-modality causal relationships following the format above.

---

## Audio-Visual Causal Pair Detection

You are an AI model specializing in multimodal reasoning. Your task is to analyze a series of short video and audio event descriptions and identify the three most challenging cross-modality relationship within them.

**Definition of "Challenging":**

A relationship is "challenging" if it meets these criteria:

**Cross-Modal Necessity**: The relationship must integrate information from both the video (visual) and audio modalities to be understood. Predicting the conclusion_event should be significantly harder or impossible using only one modality.

**Reasoning Over Perception**: The relationship should require high-level reasoning (e.g., cause-and-effect, understanding intent, diagnosing a problem, predicting social outcomes) rather than simple perception (e.g., recognizing an object, identifying a sound, describing a visible action). Avoid relationships that are merely descriptive associations.

**Diversity Consideration**: When choosing from the series, prioritize relationships that are distinct from typical or obvious ones (e.g., "person speaks" to "audio of speech"). Seek nuanced, non-obvious, or abstract connections.

**Instructions:**

Read the provided list of events.

For each event, **the modality is specified in brackets**: [V] for Video, [A] for Audio.

Identify the one relationship that best fulfills the "challenging" criteria above.

**Structure your finding exactly in the following output format:**

**1.has_causal**: [auido-video/video-audio] **premise_event_1**: [start_time_1:end_time_1][Modality]:[Event Description]. **conclusion_event_1**: [start_time_1:end_time_1][Modality]:[Event Description].

**2.has_causal**: [auido-video/video-audio] **premise_event_2**: [start_time_2:end_time_2][Modality]:[Event Description]. **conclusion_event_2**: [start_time_2:end_time_2][Modality]:[Event Description].

**3.has_causal**: [auido-video/video-audio] premise_event_3: [start_time_3:end_time_3][Modality]:[Event Description]. **conclusion_event_3**: [start_time_3:end_time_3][Modality]:[Event Description].

---

## Audio Causal Factor Scoring

You are an expert multimodal video analyst specializing in audio-visual causality. Your task is to analyze the audio channels within specific timestamps and determine the role of audio in the causal relationship between a "Premise Event" and a "Conclusion Event".

**Scoring Guide (Causal Contribution):**

**0 (None)**: The relationship is entirely visual. Removing the audio would not change a human's ability to predict the conclusion. (e.g., Walking forward → Seeing an object).

**1 (Supportive)**: Audio provides context or atmosphere that reinforces the visual link but is not strictly required to identify the conclusion. (e.g., Hearing wind reinforces why they look cold, but the visual of shivering is enough).

**2 (Essential)**: The causal link cannot be understood without the audio. The critical clue is auditory (e.g., specific speech instructions, a sound trigger like a crash prompting a reaction).

**Audio Categories:**

**Speech**: Dialogue, narration, or monologues. **Audio Event**: Environmental sounds (wind, traffic), object sounds (crashing, ringing), or physical interactions. **Music**: Background scores or diegetic music.

**Instructions:**

1. Read the `premise` and `conclusion` (ground truth). 2. Analyze the audio within the provided `cause_timestamp` and `effect_timestamp`. 3. Determine if the audio information is necessary, supportive, or irrelevant to understanding why the Premise leads to the Conclusion.

**Output Format:** Return a single JSON object with the following structure:

```
{
  "audio_types_present": {
    "speech": boolean,
    "audio_event": boolean,
    "music": boolean
  },
  "audio_description": "Brief description of what is heard...",
  "causal_contribution_score": integer (0, 1, or 2),
  "contribution_explanation": "Detailed explanation of the score."
}
```

**Time Boundary Check**

SYSTEM ERROR: TIMESTAMP VALIDATION FAILED

Your previous response contained hallucinated or logically impossible timestamps. Please correct the JSON based on the following specific errors detected:

**Detected Error:** {specific_error_msg}

**Strict Correction Rules:**

1. **No Hallucinations:** You must ONLY use timestamps that literally appear in the provided Input Caption. Do not invent time ranges (e.g., do not create "05:00-05:05" if the video ends at 04:50).

2. **Sequential Logic:** The Effect must occur *after* the Cause.

    - INVALID: Cause="02:00-02:05", Effect="02:00-02:05" (They cannot be identical).

    - VALID: Cause="02:00-02:05", Effect="02:05-02:10".

3. **Start Constraint:** As per the original instructions, the Cause Event must start AFTER 01:00 (60 seconds).

4. **Duration Limit:** The video duration is duration_string. No timestamp can exceed this limit.

**Task:**

Re-read the caption text carefully. Discard your previous invalid choice. Select a NEW Cause-Effect pair that satisfies all constraints and output the corrected JSON.

## Problem Rationality Check

You are an expert Logic Puzzle Generator specializing in Video Reasoning. Your task is to analyze a provided video caption (containing timestamps and event descriptions) and generate a single, high-quality "Next Event Prediction" multiple-choice question.

**Input Data:** You will receive raw video captions formatted as: 'Start_Time-End_Time Video: [Description] Audio: [Description]'

**Process & Constraints:**

    **1. Identify a Causal Pair (The Premise & The Effect):** Scan the text to find two events (Event A and Event B) where Event A clearly leads to or causes Event B.

        **Constraint 1 (Temporal Proximity):** The time gap between the END of Event A (Cause) and the START of Event B (Effect) must be less than 30 seconds. Ideally, it should be immediate (¡ 10s).

        **Constraint 2 (Causal Restriction):** The Premise (Event A) must logically restrict the possibilities of what happens next.

            **Bad Premise:** "John walks down the street." (Anything could happen next).

            **Good Premise:** "John trips over a crack in the pavement while holding a coffee." (Logically implies spilling, falling, or swearing).

    **2. Draft the Question:**

        **Format:** "Given the premise event: '[Description of Event A]', which event is its most direct conclusion?"

    **3. Draft the Options (1 Correct, 4 Distractors):**

        **Correct Answer:** A precise description of Event B as it appears in the caption.

        **Constraint 3 (Distractor Logic):** Distractors must be "decisive" (plausible within the scene's context) but logically inferior to the correct answer given the premise. Do not create a distractor that is a generic "common sense" outcome that is actually more likely than the video's specific outcome.

        **Distractor Types to Use:**

            Hallucinated Action: Plausible action for the character, but didn't happen.

            Wrong Object: The character interacts with a different object mentioned in the scene.

            Opposite Reaction: If the premise is sad, the distractor describes a happy reaction.

            Change Logic Alter the core logic or action of the original future event, but keep the visual objects steady. (e.g., if the original was "a man looks out the window," a changed logic could be "the man closes the curtains" – the man and window are the same objects, but the action/logic is opposite and driven by a different motivation).

            Original Counterfactual Provide a concise yet reasoned narrative of the most logical alternative future event.

            Temporal Distractor Select one event from all happened events that can mix the decision of predicting the future event.

    **5. Output Generation:**

        Return the result in the strict JSON format provided below.

**Input Caption:**

INSERT_CAPTION_HERE

**Example**

Input Caption: 00:00 - 00:05: A shot of a building at night, with one light on in a window.Modality: Video.

00:05 - 00:07: A hand presses the down arrow button on an elevator panel, and the button lights up.Modality: Video, Audio (button press).

00:07 - 00:12: The elevator doors open, and a pregnant woman walks out into a hallway.Modality: Video, Audio (elevator doors opening).

00:12 - 00:18: The woman walks into the elevator, puts her keys in her bag, and presses the button for the first floor.Modality: Video, Audio (keys jingling, button press).

00:18 - 00:22: The elevator button for the first floor lights up, and the doors begin to close.Modality: Video, Audio (elevator chime, doors closing).

00:22 - 00:25: A man's hand quickly reaches in to stop the elevator doors from closing, startling the woman.Modality: Video, Audio (sudden stop, woman gasps).

00:25 - 00:29: The man, a janitor with a mop and bucket, enters the elevator.Modality: Video, Audio (mop and bucket sounds).

00:29 - 00:32: Close-up of the janitor's boots and the mop bucket rolling into the elevator.Modality: Video, Audio (rolling sound).

00:32 - 00:36: The woman looks uncomfortable as the janitor brings his equipment into the elevator.Modality: Video.

00:36 - 00:40: The elevator doors close, trapping the woman and the janitor inside.Modality: Video, Audio (elevator doors closing).

00:40 - 00:44: The elevator begins to descend, then suddenly jolts and stops.Modality: Video, Audio (elevator moving, loud jolt, woman gasps).

00:44 - 00:47: The elevator lights flicker and go out, plunging them into darkness.Modality: Video, Audio (lights flickering, mechanical sounds).

00:47 - 00:57: The emergency lights come on, and the woman asks what happened. The janitor replies it's a "motor jam."Modality: Video, Audio (woman's voice, janitor's voice).

00:57 - 01:03: The janitor mentions being stuck in the elevator for 45 minutes last week.Modality: Video, Audio (janitor's voice).

01:03 - 01:09: The woman looks increasingly nervous.Modality: Video.

01:09 - 01:12: The elevator suddenly drops again, and the lights go out completely.Modality: Video, Audio (loud drop, woman screams, lights out).

01:12 - 01:16: The woman is terrified in the dark. The janitor says it's a "motor jam" again.Modality: Video, Audio (woman's terrified breathing, janitor's voice).

01:16 - 01:20: The janitor's hand is shown, with a fresh cut and blood.Modality: Video.

01:20 - 01:24: The janitor says he was stuck for 45 minutes last week, and the woman looks at his hand.Modality: Video, Audio (janitor's voice).

01:24 - 01:28: The janitor's hand is shown again, with more blood.Modality: Video.

01:28 - 01:32: The janitor's hand is shown gripping the mop handle, with blood on it.Modality: Video.

01:32 - 01:36: The woman looks at the janitor's hand, then down at the bloody water in the mop bucket.Modality: Video.

01:36 - 01:40: The woman looks up in horror, realizing the danger. Modality: Video.

01:40 - 01:45: The elevator display shows the numbers rapidly decreasing from 2 to 1, then the doors open. Modality: Video, Audio (elevator sounds, doors opening).

01:45 - 01:50: The woman runs out of the building and across the parking lot to her car.Modality: Video, Audio (running footsteps, woman's heavy breathing).

01:50 - 01:54: The woman fumbles with her keys, trying to unlock her car.Modality: Video, Audio (keys jingling, fumbling sounds).

01:54 - 01:58: She gets into her car and tries to start it.Modality: Video, Audio (car door closing, engine trying to start).

01:58 - 02:02: The car struggles to start, making grinding noises.Modality: Video, Audio (engine grinding).

02:02 - 02:06: The woman tries to start the car again, but it won't turn over.Modality: Video, Audio (engine grinding).

02:06 - 02:10: The woman looks frustrated and scared.Modality: Video.

02:10 - 02:14: The car's dashboard lights flicker, and the engine dies.Modality: Video, Audio (engine dying).

02:14 - 02:18: The woman looks in her rearview mirror and sees the janitor approaching.Modality: Video, Audio (woman gasps, "Shit!").

02:18 - 02:22: She frantically tries to start the car again. Modality: Video, Audio (engine grinding).

02:22 - 02:26: The car still won't start, and the woman looks terrified. Modality: Video, Audio (woman's distressed breathing).

02:26 - 02:30: The janitor is seen opening the trunk of his car. Modality: Video.

02:30 - 02:34: The woman looks back at the janitor, her face filled with fear. Modality: Video.

02:34 - 02:38: The janitor pulls out jumper cables from his trunk. Modality: Video.

02:38 - 02:42: The woman's face shows a mix of fear and confusion. Modality: Video.

02:42 - 02:46: The janitor walks towards her car with the jumper cables. Modality: Video.

02:46 - 02:50: The woman's expression changes to relief, then a slight smile. Modality: Video.

02:50 - 02:54: She tries to start the car one last time. Modality: Video, Audio (engine grinding).

02:54 - 02:58: The janitor taps on her window. Modality: Video, Audio (tap on window).

02:58 - 03:02: The janitor asks, "Need a jump?" holding up the jumper cables. The woman looks at him, a slight smile forming. Modality: Video, Audio (janitor's voice).

03:02 - 03:06: The woman smiles, relieved. Modality: Video.

03:06 - 03:10: The car finally starts. Modality: Video, Audio (car starting).

03:10 - 03:14: Title card "JUMPER" appears. Modality: Video, Audio (music starts).

03:14 - 03:38: Credits roll with animated jumper cables. Modality: Video, Audio (music continues).

{

"question": "Given the premise event: 'After the woman's car dies, the janitor is seen opening the trunk of his car', which event is its most direct conclusion?",

"options": [

"A. He retrieves a tire iron and begins walking aggressively toward the woman",

"B. He pulls out jumper cables from his trunk",

"C. He takes out the bloody mop bucket to dispose of it",

"D. He slams the trunk shut and walks back to the elevator",

"E. He removes a large flashlight and shines it into the woman's eyes"

],

"answer": "B",

"cause_timestamp": "02:26-02:30",

"effect_timestamp": "02:34-02:38",

"rationale": "The scene creates a misdirection: the woman fears the janitor is a killer (supporting distractor A or C), but the functional context is that her car has stalled (02:10). Opening the trunk is the setup for retrieving a tool to help. The visual payoff is the reveal of the jumper cables, which resolves the mechanical problem rather than the imagined horror plot. Option B is the specific event that occurs. Options A, C, and E play on the suspenseful tone but are factually incorrect."

}

---

**QA Construction**

**Role:** You are an expert in causal reasoning and narrative construction. Your task is to create plausible but incorrect effect events (distractors) that could follow a given cause event in a video. Your distractors must be deceptive, meaning they should seem reasonable at first glance but are causally invalid upon closer inspection.

**Instruction:**

1. Comprehend the Timeline: First, carefully read the entire ### TIMELINE OF EVENTS ###. Understand the sequence of actions, the characters involved, and the overall flow of the narrative. Pay attention to both visual and auditory cues. Generate four distractor effect events based on the provided cause event. You must use the specific heuristics below to guide your creation.

2. For each distractor you create, you must explicitly select and apply one of the following strategies:

1). **Reverse-Causal:** Propose an event that could have *caused* the observed cause event, effectively reversing the true temporal-causal order. (e.g., Cause: "A glass shatters." -¿ Distractor: "A ball hits the glass.").

2). **Delayed or Premature:** Propose an event that is part of the same causal chain but happens in the wrong temporal order (e.g., the consequence appears to happen before the trigger, or a later step in a sequence occurs immediately). (e.g., Cause: "A person lights a match near a firework." -¿ Distractor: "The firework explodes." [Premature: it should fizzle or fuse first]).

3). **Audio-Only Deception:** Propose an effect where the audio is highly plausible, but the visual cause is mismatched or incorrect. (e.g., Cause: "A person is chopping vegetables." -¿ Distractor Audio: "Sound of a large ceramic plate shattering." The visual would show the plate safe on the table, creating a mismatch).

4). **Video-Only Deception:** Propose an effect that is visually plausible but where the accompanying audio would contradict it, revealing the deception. (e.g., Cause: "A person is straining to lift a heavy-looking box." -¿ Distractor Video: "The box flies effortlessly into the air." The implied audio of straining is contradicted by the visual).

3.**Requirements for All Distractors:**

**Deceptiveness:** Each distractor must be a plausible continuation of the video's narrative flow.

**Causal Invalidity:** The probability of the cause event leading to the distractor must be significantly lower than the probability of it leading to the ground truth effect. The applied heuristic must create this fundamental weakness in the causal link.

**Output Format:** You must output a valid JSON object with the following structure. Do not add any other text before or after the JSON. Do not add "Video" or "Audio" text in the distractor option.

```json
{
    "cause_event": {
        "timestamp": "X",
        "event_description": "[Copy the cause event description here]"
    },
    "ground_truth_effect": {
        "timestamp": "Y",
        "event_description": "[Copy the ground truth effect description here]"
    },
    "generated_distractors": [
        {
            "event_description": "Description of distractor 1.",
```

```
            "applied_heuristic": "Name the heuristic used (e.g., Reverse-Causal, Audio-Only Deception).",
            "deceptive_rationale": "A concise explanation of how this heuristic creates a plausible but causally
invalid option."
        },
        {
            "event_description": "Description of distractor 2.",
            "applied_heuristic": "Name the heuristic used...",
            "deceptive_rationale": "A concise explanation..."
        },
        {
            "event_description": "Description of distractor 3.",
            "applied_heuristic": "Name the heuristic used...",
            "deceptive_rationale": "A concise explanation..."
        },...
    ]
} ```
```
—

Now, generate distractors for the following causal pair from a video.

---

## Error Analysis Prompt

# Role:

You are an expert analyst in Multimodal Large Language Models (MLLMs). Your task is to analyze specific failure cases from a video-audio prediction benchmark and categorize the root cause of the error.

# Input Data:

For each sample, I will provide:

1. Premise: The description of the video/audio context before the prediction.

2. Question: The question asked to the model

3. Correct Answer: The ground truth future event.

4. Model Prediction: The incorrect answer generated by the model.

5. Key Modality: The primary modality (Visual, Audio, or Both) required to solve this specific question.

# Classification Criteria:

You must classify the error into exactly one of the following four categories. Use the hierarchy below to decide:

1. Audio Perception Error:

## Definition: The correct prediction relies heavily on a specific sound (speech, event, or music) that the model clearly ignored or hallucinated.

## Indicator: The visual context alone is insufficient or misleading, and the model failed because it missed the acoustic cue (e.g., a doorbell ringing off-screen).

2. Video Perception Error:

## Definition: The correct prediction relies on a visual detail (object, text, or action) that the model failed to recognize.

## Indicator: The model's prediction contradicts clear visual evidence (e.g., predicting "driving" when the car is visually parked).

3. Lack of Knowledge:

## Definition: The model likely perceived the sensory data correctly but lacked the external world knowledge, physics, or domain expertise required to predict the outcome.

## Indicator: Understanding the scene requires prior knowledge (e.g., knowing that mixing specific chemicals causes an explosion, or knowing the rules of Chess).

4. Audio-Video Joint Reasoning Failure:

## Definition: The model correctly perceives both the visual and audio elements individually but fails to combine them logically to derive the causal future.

## Indicator: The error is not due to missing a sound or object, but failing to link them (e.g., seeing a man run + hearing a siren -¿ predicting "he is exercising" instead of "he is fleeing danger").

# Desired Output Format

Please respond in the following JSON format:

{

    "error_type": "Select one: [Audio Perception Error — Video Perception Error — Lack of Knowledge — Audio-Video Joint Reasoning Failure]",

    "reasoning": "A brief explanation of why this error falls into this category based on the difference between the prediction and the correct answer."

}

# Input Sample:

    Premise: [Insert Premise]

    Question: [Insert Question]

    Correct Answer: [Insert Correct Answer]

    Model Prediction: [Insert Model Prediction]

    Key Modality: [Insert Key Modality]

# Analysis:

---

**Video-Audio Evaluation Prompt**

These are the frames of a video and the corresponding audio.

Select the best answer to the following multiple-choice question based on the video.

Respond with only the letter (A, B, C, D, E) of the correct option.

Question: {question}

Options: {options}

---

**Audio-only Evaluation Prompt**

These are the frames of audio.

Select the best answer to the following multiple-choice question based on the audio.

Respond with only the letter (A, B, C, D, E) of the correct option.

Question: {question}

Options: {options}

---

**Video-only Evaluation Prompt**

Select the best answer to the following multiple-choice question based on the video.

Respond with only the letter (A, B, C, D, E) of the correct option.

Question: {question}

Options: {options}

