# OpenReview forum: "FutureOmni: Evaluating Future Forecasting from Omni-Modal Context for Multimodal LLMs"
_ICML.cc/2026/Conference — ICML 2026 regular_

### Official Review · Reviewer_yvbY · 2026-03-09

**Soundness:** 3
**Presentation:** 3
**Significance:** 3
**Originality:** 3
**Overall Recommendation:** 5
**Confidence:** 4

**Summary:**

This paper introduces FutureOmni, an audio-visual understanding benchmark designed to evaluate omni-modal future forecasting. FutureOmni consists of 919 videos and a total of 1,034 QA pairs. The paper conducts a comprehensive evaluation of existing multimodal LLMs. Furthermore, the authors curate a fine-tuning dataset, FutureOmni-7K, containing 7K samples. By further fine-tuning on this dataset, models can achieve performance improvements on the FutureOmni benchmark.

**Compliance With Llm Reviewing Policy:**

Affirmed.

**Final Justification:**

Regarding W1 (Model Bias), the explanation of the decoupled multi-model pipeline (Gemini, DeepSeek, and GPT-4o) alleviates my concerns about inherent model bias in the benchmark construction.

Regarding W2 (Human Verification), I think the authors have provided details of it in the rebuttal.

Regarding W3 (Reasoning Depth), I appreciate the clarification that the 30-second window does not necessarily imply shallow reasoning. The distinction between the temporal proximity of the event and the historical context required for prediction is well-taken.

Regarding W4 (OFF Strategy), the new ablation study comparing OFF with the SFT (caption-based) baseline is convincing.

Therefore, I will raise my score to this work.

---
In the Reviewer-AC discussion stage, after reading Reviewer BVma's review and his discussion with the authors, I think Reviewer BVma's concerns regarding the 'format mismatch' and 'generalization issues' are somewhat valid. However, evaluated as a benchmark paper, I personally believe this work makes a clear contribution by proposing a new task and introducing a benchmark for audio-visual future forecasting. I think this contribution is solid, and it does not warrant a Reject (score of 2). Taking into account the weaknesses raised by Reviewer BVma, I will maintain my current rating, but I will slightly lower my confidence score.

**Key Questions For Authors:**

- How the annotations for the FutureOmni-7K dataset were constructed?

**Limitations:**

yes

**Strengths And Weaknesses:**

Strengths:
- This paper focuses on an interesting task: predicting the future by combining visual and auditory cues.
- The evaluated models are comprehensive, covering 13 omni-modal and 7 video-only models, providing meaningful and informative results.

Weaknesses:
- The construction of the benchmark relies heavily on LLMs/MLLMs (e.g., Gemini 2.5 Flash). This method might bias the distribution of questions in favor of the models used for annotation, which poses potential issues. For instance, this might lead to the Gemini series models demonstrating stronger performance on the benchmark.
- Additionally, although "human verification" is mentioned in the appendix, the specific details provided are insufficient, raising concerns about the benchmark's correctness and reliability. It is strongly recommended to supplement the details of the manual verification process. Specifically, the authors should clarify how annotators evaluated each metric, what quantitative criteria were applied, and provide examples of the changes made to the manually revised samples (before vs. after modification). A detailed, accurate, and transparent human verification process is crucial for the rigorous construction of a benchmark.
- FutureOmni acts more like predicting "the next immediate event," given that the time gap between two adjacent events is strictly limited to 30 seconds. However, future forecasting should also encompass longer-term and more in-depth reasoning rather than just the immediate next action. For example, seeing a customer order and pay for a coffee at the counter, the model should forecast the barista calling their name and handing over the drink a few minutes later, rather than just predicting the immediate next event of the customer putting their wallet away.
- In Section 5, the authors propose the OFF (Omni-modal Future Forecasting) fine-tuning strategy and demonstrate that models fine-tuned on FutureOmni-7K not only perform better on the forecasting task but also exhibit improved generalization on other general benchmarks. The authors thus conclude that training on "future forecasting" enhances the model's general cross-modal capabilities. However, this conclusion might be confounded: the improvement could merely stem from the higher quality and richer diversity of the videos present in FutureOmni-7K, rather than the specific impact of "learning the future forecasting task itself."

---

> ### Author Rebuttal · Authors · 2026-03-29
>
> We sincerely thank the reviewer for this highly constructive questions.
>
> ---
>
> > `W1`: "About model biases"
>
> A: We anticipated this risk during the design of FutureOmni and intentionally implemented a Decoupled, Multi-Model Pipeline to mitigate this exact issue. It is important to clarify that Gemini 2.5 Flash was not responsible for creating the questions or the causal logic. Gemini 2.5 Flash was used strictly as a high-fidelity perceptual tool for Audio-Visual Temporal Localization and
> Calibration (Section 3.2)—objectively extracting dense, timestamped visual and acoustic events. The actual "intelligence" of the benchmark—the Causal Pair Discovery and QA Construction (Section 3.3)—was driven by an entirely different model family: DeepSeek-V3.2. Furthermore, the automated logical validation step utilized a third distinct architecture: GPT-4o.
>
> ---
>
> > `W2`: "About human verification details"
>
> A: Our human verification was conducted by a team of 2 expert annotators. Every candidate QA pair that passed the automated GPT-4o check (and specifically all high-difficulty pairs with an Audio Contribution Score of 2) underwent independent manual review.
> Annotators evaluated each sample based on three criteria:
> - Causal Strictness (1-3 scale): Does the premise unambiguously and logically lead to the correct future event? (1: Ambiguous/Weak link, 2: Plausible but has alternatives, 3: Strict and necessary logical conclusion). Only samples scoring 3 were accepted.
> - Audio-Visual Necessity (Binary - Yes/No): Is it impossible to answer correctly using only the video frames or only the audio track? If "No", the sample was rejected or sent for revision.
> - Distractor Validity (1-3 scale): Are the distractors (Visual-only, Audio-only, Delayed, Reverse-Causal) plausible but objectively incorrect based on the omni-modal context?. Only sample scoring 2 of all distractors were accepted.The revised example is showed in the following link: https://anonymous.4open.science/r/FutureOmni_anon-1F01/rebuttal/revised_example.png.
>
> ---
>
> > `W3`: "About the in-depth reasoning of future forecasting"
>
> We respectfully clarify that a strict 30-second temporal gap between the trigger and the event does not equate to shallow, immediate-action prediction. As detailed in Section 3.3, our pipeline explicitly filters out simple "immediate kinematic consequences" (e.g., predicting a dropped glass will shatter, or a customer putting away a wallet). These rely purely on basic visual physics and common sense, lacking power for omni-modal reasoning. Our human verification rejected such trivial samples. Crucially, while the gap between cause and effect is short (to ensure an unambiguous ground-truth), the reasoning required often relies on long-term historical context established minutes earlier. The Tom and Jerry example in Figure 1 illustrates this. The premise (trigger) is Tom stepping on tacks. The "common sense" immediate event is screaming. However, the true future event (Tom enduring the pain silently and running outside) requires the model to remember a critical acoustic constraint established minutes prior: the King threatened him if he made noise.
>
> ---
>
> > `W4`: "About cofounded conclusion of OFF"
>
> A: In following table, we introduced a baseline named "SFT", where the models were fine-tuned using video captions generated at first step in Section 3.2.
> The empirical results explicitly demonstrate the unique superiority of our "future forecasting" strategy:
> 1. Consistent Superiority Across Models & Benchmarks: Across three different base models (Qwen2.5-Omni, video SALMONN 2, Ola) and five diverse benchmarks, OFF consistently outperforms the generic SFT baseline.
> 2. Significant Gains in Temporal/Predictive Tasks: The difference is particularly pronounced in benchmarks requiring temporal reasoning and anticipation, such as FutureOmni. For video SALMONN 2, our forecasting data achieves a performance of 49.90, significantly surpassing the caption baseline of 46.89 by a large margin.
>
> | Model | FutureOmni | | WorldSense | | DailyOmni | | JointAVBench | | OmniVideoBench | |
> | :--- | :---: | :---: | :---: | :---: | :---: | :---: | :---: | :---: | :---: | :---: |
> | | OFF | SFT | OFF | SFT | OFF | SFT | OFF | SFT | OFF | SFT |
> | Qwen2.5-Omni | 48.51 | 47.09 | 40.22 | 38.49 | 49.03 | 47.70 | 60.88 | 59.58 | 31.70 | 31.20 |
> | video SALMONN2 | 49.90 | 46.89 | 48.77 | 48.01 | 65.80 | 64.97 | 61.16 | 60.43 | 35.40 | 34.33 |
> | Ola | 50.19 | 48.02 | 44.19 | 44.01 | 53.63 | 53.05 | 51.13 | 50.11 | 36.90 | 35.09 |
>
> ---
>
> > `Q1`: "The annotation details of FutureOmni-7K"
>
> A: The FutureOmni-7K instruction-tuning dataset was constructed using the exact same robust pipeline developed for the evaluation benchmark, ensuring high-quality, omni-modal causal dependencies. The training videos and testing videos have no overlap. The characteristic of FutureOmni-7K is the inclusion of the Rationale.

---

> > ### Author Rebuttal · Reviewer_yvbY · 2026-04-02
> >
> > Thank you for the detailed rebuttal. My concerns are addressed.

---

### Official Review · Reviewer_BVma · 2026-03-10

**Soundness:** 2
**Presentation:** 2
**Significance:** 3
**Originality:** 2
**Overall Recommendation:** 2
**Confidence:** 5

**Summary:**

The authors present FutureOmni, a new benchmark designed to evaluate the capability of Multimodal Large Language Models (MLLMs) to perform future event forecasting using both visual and auditory inputs. Unlike existing benchmarks that focus primarily on retrospective reasoning (describing what has already happened) or vision-centric prediction, FutureOmni emphasizes omni-modal causal and temporal reasoning. The dataset consists of 919 videos and 1,034 multiple-choice QA pairs across 8 domains. The authors also propose the Omni-Modal Future Forecasting (OFF) instruction-tuning strategy and provide a 7K-sample dataset to improve model performance.

**Compliance With Llm Reviewing Policy:**

Affirmed.

**Final Justification:**

I have read the rebuttal and follow-up discussions. My score remains 2.

The authors' responses do not resolve my core concerns:
- OFF remains standard supervised fine-tuning with a reformulated target format, reframing this as a "paradigm shift" does not constitute methodological novelty.
- The caption-style SFT baseline is not a fair comparison due to fundamental format mismatch.
- The modest, inconsistent improvements on broader benchmarks fail to support the generalization claims.

The benchmark contribution is appreciated, but insufficient to overcome these weaknesses.

**Key Questions For Authors:**

- The proposed "OFF" method is a standard instruction-tuning pipeline. Beyond dataset creation, what is the algorithmic innovation here?
- Can the authors provide a more rigorous, quantitative analysis to explain why future-forecasting training transfers to retrospective benchmarks, beyond the descriptive attention score?

**Limitations:**

- The authors acknowledge that the proposed "OFF" method is essentially a supervised fine-tuning strategy. They have not explored whether the observed performance gains are inherent to their specific "future forecasting" data or if a comparable performance improvement could be achieved via a generic instruction-tuning dataset of similar scale.

**Strengths And Weaknesses:**

Strengths:

- The paper identifies a under-explored area in multimodal research, the shift from retrospective understanding (describing past events) to predictive reasoning (forecasting future events) within an omni-modal context (audio-visual).
- The authors have clearly put substantial effort into the creation of the FutureOmni benchmark.
- The design of the multiple-choice QA pairs is commendable, particularly the inclusion of various distractor types (e.g., visual-only perception, audio-only perception, and reverse-causal distractors).
- The evaluation is extensive, spanning 20 representative models (both proprietary and open-source).

Weaknesses:

- The proposed solution, "OFF" (Omni-Modal Future Forecasting), lacks significant methodological novelty. It is essentially a standard supervised instruction-tuning process .
- The paper is heavily skewed towards dataset and benchmark engineering.

---

> ### Author Rebuttal · Authors · 2026-03-29
>
> We sincerely thank the reviewer for this highly constructive questions.
>
> ---
>
> > `Q1&W1&W2`: "The algorithmic innovation of OFF"
>
> A: We respectfully argue that our innovations lie in in the paradigm shift of the training objective and the structural redesign of the supervision signals. Specifically, standard video instruction-tuning typically maps a complete input video directly to a descriptive output (e.g., Video $\rightarrow$ Caption). The OFF strategy restructures this mapping to enforce **causal reasoning**. Instead of learning $P(Answer\mid Video, Question)$, OFF forces the model to learn $P(Rationale,Answer\mid Video_{t\leq t_{split}},Premise)$. $t_{split}$ denotes the split point based on the causal event and $Video_{t\leq t_{split}}$ represents the truncated video before the split point. By explicitly injecting the causal reasoning chain into the training target, **OFF transforms the task from simple visual-semantic matching into logical deduction (causal and temporal reasoning).** Moreover, this work does not primarily aim to propose a novel algorithm. Instead, we focus on introducing a **new omni-modal future forecasting task**, constructing a high-quality training and evaluation benchmark, and providing a comprehensive empirical study over 20+ omni-modal large models.
>
> ---
>
> > `Q2`: "About explaining why future-forecasting transfers"
>
> A: We argue that OFF improves transferability by explicitly training models to identify causal triggers and reason over temporal progression. This encourages the model to develop a structured understanding of timelines, rather than relying on shallow correlations. The following table supports our hypothesis by comparing the accuracy gains ($\Delta Acc =  Acc_{train} - Acc_{original}$) of Qwen2.5-Omni trained with OFF versus standard SFT with video captions across fine-grained temporal tasks (WorldSense: Temporal Prediction, Temporal Localization, JointAVBench: Plot Temporal Grounding, OmniVideoBench: Temporal Understanding). By explicitly breaking down these performance gains, we quantitatively demonstrate that the transferability of the OFF strategy is not a superficial, generalized boost. Instead, it is a targeted enhancement of the precise cognitive skill—**Temporal Reasoning**—demanded by future prediction. For completeness, we also provide the full task-level improvements here:: https://anonymous.4open.science/r/FutureOmni_anon-1F01/rebuttal/Improvements_over_3_benchmarks.png.
>
>
> | Benchmark          | Fine-grained Task Type                          | $\Delta Acc$ |      |
> | ------------------ | ----------------------------------------------- | ---------------- | ----- |
> |                    |                                                 | **OFF**          | **SFT** |
> | **WorldSense**     | Temporal Localization                        | **1.78%**  | -0.59%       |
> |                    |    Temporal Prediction                              |  **5.45%** | 1.82%       |
> | **JointAVBench**   | PTG *(Plot Temporal Grounding)*  | **6.62%** | 2.21%       |
> | **OmniVideoBench** | Temporal Understanding      | **2.92%** | 1.46%      |
>
> ---
>
> > `L1`: "About whether performance improvements can be achieved by a generic SFT dataset"
>
> A: In following table, we introduced a baseline named "SFT", where the models were fine-tuned using  video captions of a  comparable scale to  FutureOmni-7K, generated at first step in Secton3.2.
> The empirical results explicitly demonstrate the superiority of our OFF:
> 1. **Consistent Superiority Across Models & Benchmarks**: Across three base models (Qwen2.5-Omni, video SALMONN 2, Ola) and five benchmarks, OFF consistently outperforms the generic SFT baseline.
> 2. **Significant Gains in Temporal/Predictive Tasks**: The difference is pronounced in benchmarks requiring temporal reasoning and anticipation, such as FutureOmni. For video SALMONN 2, our forecasting data achieves a performance of 49.90, surpassing the caption baseline of 46.89 by a large margin. Similar trends are observed in the Ola (50.19 vs. 48.02).
>
> **Conclusion:**
> These results firmly indicate that the observed improvements are not merely a byproduct of  training data. Instead, they are inherent to the specific nature of our "future forecasting" task, which forces the model to internalize causal relationships and temporal progression—capabilities that standard descriptive generic datasets (SFT) fail to impart.
>
> | Model | FutureOmni | | WorldSense | | DailyOmni | | JointAVBench | | OmniVideoBench | |
> | :--- | :---: | :---: | :---: | :---: | :---: | :---: | :---: | :---: | :---: | :---: |
> | | OFF | SFT | OFF | SFT | OFF | SFT | OFF | SFT | OFF | SFT |
> | Qwen2.5-Omni | 48.51 | 47.09 | 40.22 | 38.49 | 49.03 | 47.70 | 60.88 | 59.58 | 31.70 | 31.20 |
> | video SALMONN2 | 49.90 | 46.89 | 48.77 | 48.01 | 65.80 | 64.97 | 61.16 | 60.43 | 35.40 | 34.33 |
> | Ola | 50.19 | 48.02 | 44.19 | 44.01 | 53.63 | 53.05 | 51.13 | 50.11 | 36.90 | 35.09 |

---

> > ### Author Rebuttal · Reviewer_BVma · 2026-04-03
> >
> > Thank you for the rebuttal. It does not change my assessment.
> >
> > - My main concern is **originality**. **OFF** is, in substance, **standard supervised instruction tuning** with a forecasting-oriented target format. This is a **data/task reformulation**, not a substantive methodological innovation.
> >
> > - The new **caption-style SFT** comparison does not resolve this point because it is **not a fair baseline**. **OFF** is trained directly in an **MCQA** format that is much closer to the downstream evaluation benchmarks, **all of which are MCQA**, whereas **caption-style SFT** is fundamentally mismatched in **objective** and **output format**. As such, this comparison cannot support a distinct method claim: it mainly shows that **task-matched supervision outperforms generic caption tuning**, which is expected.
> >
> > - Further, **Table 5 in the paper fails to provide convincing evidence of broad generalization**: improvements on broader benchmarks are largely **small and inconsistent**, and some benchmarks even show **slight drops**. This **substantially weakens the claim** that the proposed **training signal** yields a capability that transfers to **general domains**.
> >
> > For these reasons, I keep my original score.

---

> > > ### Author Response · Authors · 2026-04-03
> > >
> > > We sincerely thank the reviewer for the continued engagement.
> > >
> > > ---
> > >
> > > > `1`: On OFF Method Originality
> > >
> > > A: We would like to clarify that **we do not position this work as proposing a novel model architecture.** Instead, our contribution lies in **problem formulation, benchmark design, and training paradigm.**
> > > As stated in Lines 100–109, our contributions are:
> > >
> > > (1) We introduce the **first omni-modal future forecasting task and dataset**, requiring prediction under partial observability.
> > >
> > > (2) We provide a **systematic evaluation of 20+ open- and closed-source MLLMs**, revealing consistent limitations in causal-temporal reasoning.
> > >
> > > (3) We construct a **7K instruction-tuning dataset** and propose the OFF training strategy, which reformulates supervision to enforce causal-temporal reasoning.
> > >
> > > Specifically, OFF is a **training-time reformulation** that enforces **input truncation at causal trigger points**, preventing access to future frames and forcing **forward extrapolation** rather than retrospective summarization.
> > >
> > > ---
> > >
> > > > `2`: On the “format matching” baseline concern
> > >
> > > A: We agree that a fully task-matched baseline would be ideal. In our current setup, the SFT baseline is constructed from the same **7K dataset** using QA-style caption supervision, ensuring **identical video exposure** and isolating the effect of supervision structure.
> > >
> > > We have considered the following stronger alternatives:
> > >
> > > (1) **Fully matched MCQA SFT on our 7K dataset**, which would require large-scale human annotation for high-quality distractors and question design, making it prohibitively costly.
> > >
> > > (2) **Pseudo-MCQA from captions**, which remains descriptive rather than predictive. Moreover, captions are typically much longer than forecasting targets, introducing significant distribution mismatch; thus, it fails to capture the forecasting objective.
> > >
> > > (3) **External omni-modal MCQA datasets**, which are not available at a comparable scale and differ substantially in data distribution, video source, and task design, making them not directly comparable.
> > >
> > > Given these constraints, we adopt caption-style SFT as a controlled and practical baseline, particularly to **ensure identical video source**. **We would greatly appreciate any more specific suggestions on the baseline design and are happy to explore them.**
> > >
> > > ---
> > >
> > > > `3`: On “format alignment explains the gains”
> > >
> > > A: We agree that caption-style SFT is not perfectly aligned with the downstream format. However, our conclusions do **not rely solely on forecasting tasks.**
> > >
> > > Importantly, OFF consistently improves performance on multiple retrospective benchmarks (WorldSense, JointAVBench, OmniVideoBench, MLVU), where:
> > > (1) the tasks are **not forecasting**,
> > > (2) **no rationale** is involved, and
> > > (3) there is **no alignment in output format or reasoning structure.**
> > > If the gains were solely due to format matching, they would not transfer to these out of domain benchmarks.
> > >
> > > We hope our response addresses your concerns. We would be happy to further explore any additional questions or suggestions and look forward to your feedback.
> > >
> > > Best regards,
> > >
> > > Authors of #9960

---

### Official Review · Reviewer_bNnG · 2026-03-15

**Soundness:** 3
**Presentation:** 3
**Significance:** 2
**Originality:** 2
**Overall Recommendation:** 4
**Confidence:** 3

**Summary:**

This paper introduces FutureOmni, the first benchmark designed to evaluate the capability of multimodal LLMs to predict future events derived from integrated audio-visual contexts. The dataset comprises 919 videos and 1034 multiple-choice QA pairs spanning 8 domains. To prevent shortcut learning, the authors incorporate 4 categories of adversarial distractors (vision-only, audio-only, delayed, and anti-causal). Evaluation of 20 MLLMs indicates that the top-performing model, Gemini 3 Flash, achieves an accuracy of only 64.8%. The paper also introduces the OFF training strategy alongside a 7K instruction-tuning dataset.

**Compliance With Llm Reviewing Policy:**

Affirmed.

**Final Justification:**

I maintain my positive rating.

My initial concerns focused on several aspects of the experimental design and evaluation methodology. The authors provided a thorough rebuttal that adequately addressed these concerns, with convincing clarifications and additional experimental evidence that strengthened the empirical foundation of the work.

Overall, the paper makes a timely and well-motivated contribution by identifying an underexplored yet important capability gap in current multimodal LLMs. The benchmark is well-constructed with meaningful adversarial design choices. I encourage the authors to incorporate the additional results from the rebuttal into the final manuscript.

**Key Questions For Authors:**

1. Could the authors provide an ablation study (comparing performance with and without the audio track) to quantitatively isolate the contribution of the audio modality to the predictive accuracy?
2. To what extent might models be exploiting the multiple-choice format via elimination heuristics? How do the top-performing models fare when evaluated in an open-ended generative setting?
3. Does the highly variable temporal span of the "future prediction" targets (ranging from near-instantaneous reactions to 30-second horizons) negatively impact the internal consistency and reliability of the evaluation?

**Limitations:**

yes

**Strengths And Weaknesses:**

### Strengths
1. **Fills a distinct research gap:** While existing benchmarks heavily index on retrospective comprehension, the intersection of predictive forecasting and the audio modality remains a largely unexplored paradigm.
2. **Rigorous data construction pipeline:** The curation workflow—ranging from audio-coordinated video filtering and MFCC temporal alignment to causal pair discovery and dual-stage validation (GPT-4o + human annotators)—is exceptionally thorough.
3. **Effective adversarial distractor design:** The inclusion of "vision-only" and "audio-only" distractors serves as a highly effective probe to determine if models are executing genuine cross-modal reasoning rather than relying on unimodal priors.
4. **Large-scale evaluation:** The benchmark comprehensively evaluates 20 distinct MLLMs (13 omni-modal and 7 video-only architectures), encompassing both open-weights and proprietary systems.
5. **Empirical efficacy of the OFF method:** The proposed training strategy demonstrates measurable performance gains not only on FutureOmni but also across WorldSense, DailyOmni, and Video-MME, suggesting robust generalization properties.

### Weaknesses
1. **Constrained benchmark scale:** A dataset of 1034 QA pairs is relatively small when juxtaposed with established benchmarks in the space (e.g., WorldSense at 3172, VLEP at 4192).
2. **Limitations inherent to multiple-choice formats:** The multiple-choice structure introduces the risk of models achieving artificially high scores via process-of-elimination heuristics. The paper would benefit significantly from the inclusion of an open-ended generative evaluation track.
3. **Homogeneous data sourcing:** The video corpus is sourced predominantly from YouTube, resulting in a systemic underrepresentation of other critical domains such as cinematic footage, animation, and static surveillance.
4. **Incremental novelty of the OFF method:** The proposed training methodology is functionally standard instruction tuning combined with rigorous data filtering; it does not introduce a fundamentally novel training paradigm.
5. **Ambiguous definition of "future forecasting":** The temporal span of the predictive targets is highly inconsistent. Some QA pairs require predicting immediate kinematic consequences just seconds away, while others demand complex, longer-term causal forecasting.

---

> ### Author Rebuttal · Authors · 2026-03-29
>
> We sincerely thank for the suggestions.
>
> ---
>
> > `W1&W3`: "Limited scale and homogeneous video data"
>
> A: While our 1,034 QA pairs are fewer than VLEP or WorldSense, our benchmark is more rigorous:
> - **Near a 1:1 Video-to-QA Ratio**: To prevent models from exploiting intra-video correlations, we strictly maintain 1,034 QAs across 919 unique videos. In contrast, WorldSense inflates its 3,172 QAs by repeatedly querying the same 1,662 videos.
> - **Task Complexity**: FutureOmni demands long-context reasoning (Avg. 163.5s duration) with 5 adversarial options designed to punish unimodal shortcuts. Conversely, VLEP relies on very short clips (Avg. 33.1s) with simple binary (2-option) choices, which are highly susceptible to guessing heuristics.
>
> We respectfully disagree that sourcing from YouTube leads to homogeneity. As demonstrated by flagship benchmarks like Video-MME, YouTube is a massively heterogeneous aggregation platform.
>
> ---
>
> > `W2&Q2`: "About multiple-choice format"
>
> A: The fundamental challenge of "future forecasting" is its inherent uncertainty; without constraints, multiple futures could be deemed plausible. The MCQ format in FutureOmni is a deliberate design choice to provide necessary narrative boundaries.
> Crucially, our MCQ design actively punishes elimination heuristics. As detailed in Section 3.3, we do not use random false options. We engineered four adversarial distractor types (Visual-Only, Audio-Only, Delayed, Reverse-Causal) in Section 3.3.  These options are highly context-relevant, meaning a model cannot eliminate them based on "topic mismatch." It must genuinely synthesize audio-visual causal logic to reject them.
> For open-ended, we actually conducted an additional evaluation on a representative subset of FutureOmni (100 complex causal and thematic samples) using the top-performing models: Gemini 3 Flash, GPT-4o, and Qwen3-Omni.
> We provided the models with the video/audio up to the split point and simply prompted: "Describe in detail the most logical immediate next event."   We evaluated the responses using LLM-as-a-Judge (Claude Sonnet 4.6 and Grok 4.2) scoring protocol (1-5 scale) focusing on causal consistency and audio-visual integration.  The results in following table showed that:  i) without the constraints of MCQ options, all models experienced a significant performance drop. ii) the variance between different LLM judges demonstrates that automated open-ended evaluation for multimodal forecasting remains highly subjective and unreliable.
>
> | Evaluated Model   | Score by Grok 4.2 | Score by Claude Sonnet 4.6  |
> |-------------------|---------------------------|-------------------------------------|
> | GPT-4o            |      2.2           |    2.1                       |
> | Gemini 3 Flash    | 2.7               |     2.9                      |
> | Qwen3-Omni      |    1.9             | 2.2                          |
>
> ---
>
> > `W5&Q3`:"About the temporal span"
>
> A: Firstly we argue that our span with 30 seconds is not large since other related benchmarks (FutureX, FutureBench) are based on minutes and days. And we actively designed our pipeline (Section 3.3) to filter out trivial "immediate kinematic consequences" (e.g., a dropped glass falling). These rely purely on basic visual physics and lack discriminative power for evaluating omni-modal reasoning. Real-world videos (movies, documentaries) are not continuous spatial-temporal streams. Due to camera cuts and scene transitions, a cause and its effect are rarely perfectly adjacent. A 30-second window is necessary to accommodate these natural cinematic leaps. For evaluation reliability, our unifying metric is Causal Determinacy, not temporal proximity. Our Dual-Stage Verification rigorously ensures that whether the target event occurs 5 or 25 seconds later, the antecedent premise provides unambiguous audio-visual evidence to logically infer the singular outcome.
>
> ---
>
> > `W4`: "About novelty of OFF"
>
> We respectfully argue that our innovations lie in in the paradigm shift of the training objective and the structural redesign of the supervision signals. Specifically, standard video instruction-tuning typically maps a complete input video directly to a descriptive output (e.g., Video $\rightarrow$ Caption). The OFF strategy restructures this mapping to enforce causal reasoning. Instead of learning $P(Answer\mid Video, Question)$, OFF forces the model to learn $P(Rationale,Answer\mid Video_{t\leq t_{split}},Premise)$. $t_{split}$ denotes the split point based on the causal event and $Video_{t\leq t_{split}}$ represents the truncated video before the split point. By explicitly injecting the causal reasoning chain into the training target, OFF transforms the task from simple visual-semantic matching into logical deduction.
>
> ---
>
> > `Q1`: About modality ablation
>
> We are pleased to inform the reviewer that we have already conducted a comprehensive Modality Ablation study, which is detailed in Table 3 of our manuscript.

---

> > ### Author Rebuttal · Reviewer_bNnG · 2026-04-03
> >
> > I thank the authors for their thorough response, which has addressed most of my concerns. I am satisfied with the additional experiments and clarifications provided. I will maintain my original positive score.

---

### Official Review · Reviewer_BYqJ · 2026-03-17

**Soundness:** 3
**Presentation:** 4
**Significance:** 3
**Originality:** 2
**Overall Recommendation:** 4
**Confidence:** 4

**Summary:**

This paper introduces FutureOmni, a novel benchmark designed to evaluate the omni-modal future forecasting capabilities of Multimodal Large Language Models. While previous benchmarks have largely focused on retrospective understanding FutureOmni challenges models to predict future events by synthesizing both visual and auditory cues. To support this task, the authors developed the FutureOmni-7K instruction-tuning dataset and proposed the OFF training strategy. Results demonstrate that the OFF strategy not only bolsters future prediction performance but also enhances general video understanding and generalization.

**Compliance With Llm Reviewing Policy:**

Affirmed.

**Key Questions For Authors:**

1. Since specific LLM models were utilized in the dataset construction process, what strategies were employed to identify and mitigate potential model-specific biases in the generated data?

2. Could you provide a clearer distinction regarding the specific strengths and weaknesses of the OFF strategy in comparison to standard multi-modal instruction tuning techniques?

3. During the benchmark generation process, did you identify any unique patterns or logical constraints in the AI-generated questions compared to those produced by human annotators? Additional analysis regarding how these questions maintain human-level validity would be highly valuable.

**Limitations:**

yes

**Strengths And Weaknesses:**

The primary strength of this work lies in its expansion of future forecasting from a vision-only scope to an integrated audio-visual perspective, which more accurately captures the causal complexity of real-world environments. The implementation of four types of adversarial distractors is a commendable design choice that effectively mitigates the risk of shortcut learning. Furthermore, the hybrid pipeline (combining automated LLM-based annotation with human-in-the-loop verification) demonstrates a sophisticated approach to maintaining high data quality while ensuring scalability.
On the other hand, the reliance on specific LLM architectures during the dataset construction process introduces a potential risk of model bias. The current scale of the dataset is somewhat limited, suggesting that expansion into more diverse domains would be beneficial. While the authors demonstrate that the OFF strategy improves performance, absolute accuracy remains relatively low, and the paper would have been strengthened by a deeper analysis of the fundamental barriers to achieving higher performance in these tasks.

---

> ### Author Rebuttal · Authors · 2026-03-29
>
> We sincerely thank the reviewer for highlighting this critical point.
>
> ---
>
> > `W1& Q1`: "How to mitigate model-specific biases"
>
> A: To  mitigate biases in FutureOmni, we implemented :
> - Strict Prompt Constraints (Mitigating Stylistic Bias): During the Causal Pair Discovery and QA Construction phases (Section 3.3), we explicitly constrained DeepSeek-V3.2 to generate concise, objective event descriptions. This ensured the options remained neutral and grounded purely in the visual and acoustic evidence.
> - Dual-Stage Verification : We employed GPT-4o—as an independent logical validator in the first stage of our verification process. Candidate QA pairs  were evaluated by GPT-4o for logical strictness, uniqueness of the correct answer, and the validity of distractors. And expert annotators manually reviewed the video, audio, and generated QA pairs to filter out hallucinated correlations, illogical distractors, and any residual model-specific quirks. We quantified this process with an Inter-Annotator Agreement (IAA) score, demonstrating the high quality and human-aligned logic of the final benchmark.
>
> ---
>
> > `W2`: "Scale is limited"
>
> A: Current scale was a deliberate design. Generating QAs where future is determined by synergistic visual and acoustic cues required our complex pipeline followed by human verification. We choose to release a tightly curated benchmark rather than a massive but noisy dataset. And FutureOmni is featured with 8 domains with 21 fine-grained categories.
>
> ---
>
> > `W3`: "Analysis for barriers to achieve high performance "
>
> A: From Figure 6, we conducted an error analysis and found that video perception and audio-video
> joint reasoning error are main root cause for prediction failure (See in Section4, Error Analysis).
>
> ---
>
> > `Q2`: "the specific strengths and weaknesses of the OFF"
>
> A: Strength: Since OFF forces the model to pinpoint exact causal "triggers" and extrapolate timelines, we argue that OFF   teaches the models how to reason over timelines. The following table supports our hypothesis by comparing the accuracy gains ($\Delta Acc=Acc_{train} - Acc_{original}$) of Qwen2.5-Omni trained with OFF versus SFT with video captions  across fine-grained temporal tasks. The table of accuracy gains for whole tasks is listed in following link: https://anonymous.4open.science/r/FutureOmni_anon-1F01/rebuttal/Improvements_over_3_benchmarks.png.
>
> Weakness: We also acknowledge that constructing high-quality OFF data requires a complex, multi-model pipeline (Gemini + DeepSeek) and human-in-the-loop verification to ensure strict causal determinacy and valid adversarial distractors. This makes scaling OFF datasets significantly more expensive and labor-intensive.
>
> | Benchmark          | Fine-grained Task Type                          | $\Delta Acc$ |      |
> | ------------------ | ----------------------------------------------- | ---------------- | ----- |
> |                    |                                                 | **OFF**          | **SFT** |
> | **WorldSense**     | Temporal Localization                        | **1.78%**  | -0.59%       |
> |                    |    Temporal Prediction                              |  **5.45%** | 1.82%       |
> | **JointAVBench**   | PTG *(Plot Temporal Grounding)*  | **6.62%** | 2.21%       |
> | **OmniVideoBench** | Temporal Understanding      | **2.92%** | 1.46%      |
>
> ---
>
> > `Q3`: "Special patterns & Human validation"
>
> A: During the construction of FutureOmni, we did indeed observe distinct patterns and logical constraints.
> - Hyper-Granularity vs. Abstraction: AI models tend to focus on immediate physical consequences (e.g., "The glass hits the floor and shatters") but struggle with abstract or socially nuanced predictions (e.g., "The person looks embarrassed").
> - The "Safe Distractor" Bias: AI frequently generated logically impossible or completely irrelevant distractors, rendering multiple-choice questions too easy and allowing models to guess via simple semantic matching rather than true causal forecasting.
> - Unimodal Leakage: When generating omni-modal questions, the AI initially exhibited visual dominance, often creating questions where the audio context was decorative rather than causally necessary.
>
> To ensure the final FutureOmni benchmark  maintains human-level validity, we explicitly designed:
> - Adversarial Distractor Engineering (Section 3.3): we rigidly constrained the prompt to produce four specific types of adversarial distractors: Visual-only Perception, Audio-only Perception, Delayed, and Reverse-Causal.
> - Forcing Acoustic Causality and verification (Section ): To mitigate unimodal leakage, we introduced the "Audio Causal Factor Scoring" step. We explicitly required the AI to score (0-2) the necessity of the audio cue. Only pairs where audio was deemed a decisive causal factor (Score = 2) were prioritized for the final benchmark.

---

> > ### Author Rebuttal · Reviewer_BYqJ · 2026-04-05
> >
> > Thank you for the rebuttal, I will keep my original score.

---

### Decision · Program_Chairs · 2026-04-30

**Decision:**

Accept (regular)

**Comment:**

This paper receives mixed reviews: Weak accept, weak accept, reject, accept. All reviewers appreciate the contribution of this paper by introducing a new benchmark for an interesting problem (i.e., predicting the future by combing visual and audio cues). The only negative reviewer raises concern on the technical novelty of still using SFT, potentially unfair comparison, and inconsistent improvement. For the novelty issue, the paper mainly focuses on defining a new task and building new benchmark. Thus, it is acceptable to use a standard SFT method to demonstrate the performance of baseline methods. For the unfair comparison, the format mismatch might be a practical issue and the rebuttal further provides new results to justify its design. For the generalization claims, the rebuttal also provides new results to illustrate that it can explicitly enable training models to identify causal triggers and reason over temporal progression. Overall, the AC agrees with the majority of the reviewers, and think this work makes a clear contribution by proposing a new task and introducing a benchmark for audio-visual future forecasting.  The AC makes an accept recommendation. The authors are encouraged to include these discussions in the rebuttal into the final version.